# CaP-X: A Framework for Benchmarking and Improving Coding Agents for Robot Manipulation

Letian Fu [* 1 2]  Justin Yu [* 2]  Karim El-Refai [* 2]  Ethan Kou [* 2]  Haoru Xue [* 1 2]  Huang Huang [3]  Wenli Xiao [4]
Fei-Fei Li [3]  Guanya Shi [4]  Jiajun Wu [3]  Shankar Sastry [2]  Yuke Zhu [1]
Ken Goldberg [† 2]  Linxi "Jim" Fan [† 1]

## Abstract

Code-as-Policy (CaP) is a paradigm in which a language or vision-language model generates executable robot control programs, yet its effectiveness as an autonomous controller for embodied manipulation remains underexplored. Prior CaP systems often rely on high-level, human-designed primitives, making it difficult to separate agent capability from designer-provided scaffolding. We present **CaP-X**, an open-access framework for systematically studying Code-as-Policy agents in robot manipulation. CaP-X includes four components. **CaP-Gym** is an interactive environment in which coding agents control robots by synthesizing and executing programs that compose perception and control primitives. Building on this foundation, **CaP-Bench** evaluates frontier language and vision-language models across varying levels of abstraction, interaction, and perceptual grounding. Across 12 models, the task success rates improve with human-crafted abstractions but degrade as these priors are removed, exposing a dependence on designer scaffolding. At the same time, we observe that scaling test-time computation with multi-turn interaction, structured execution feedback, visual differencing, automatic skill synthesis, and ensembled reasoning can substantially improve robustness even when agents operate over low-level primitives. These findings motivate **CaP-Agent0**, a training-free framework that achieves near human-level reliability on several manipulation tasks in simulation and on real embodiments. **CaP-RL** explores reinforcement learning with verifiable rewards to improve success rates and supports sim-to-real

transfer through a shared code-as-action-space interface. Together, CaP-X provides an open-access platform for advancing embodied coding agents. Project page: https://capgym.github.io

## 1. Introduction

Robots have long been controlled through explicit programs that combine perception, geometry, planning, and feedback (Fikes & Nilsson, 1971; Murray et al., 1994; Aeronautiques et al., 1998; Siciliano et al., 2008). As robots advanced to continuous, higher-dimensional spaces, these representations were integrated with geometric motion planning (Khatib, 1986), evolving into Task-and-Motion Planning (TAMP) (Kaelbling & Lozano-Pérez, 2011). These *classical* control paradigms achieve robustness through explicit structure: human engineers manually write software that decomposes high-level goals into subtasks, compose perception and control modules, and handle failure and edge cases through trial-and-error and explicit logic. While this can offer strong interpretability and geometric precision guarantees, it relies heavily on human skill and expertise. The manual process can be very time-consuming and often produces task-specific solutions that are difficult to generalize to open-ended environments.

Driven by successes in foundation models (Devlin et al., 2018; Radford et al., 2018; 2019; Brown et al., 2020; Chowdhery et al., 2023; Achiam et al., 2023; Radford et al., 2021; Li et al., 2023), another robot control paradigm has emerged in the form of Vision-Language-Action (VLA) models (Brohan et al., 2023; Kim et al., 2024; Octo Model Team et al., 2024; Jang et al., 2022; Jiang et al., 2023; Reed et al., 2022; Open X-Embodiment Collaboration et al., 2024; Shah et al., 2023; Fu et al., 2024; Huang et al., 2025; Bjorck et al., 2025; TRI LBM Team et al., 2025; Physical Intelligence et al., 2025). These approaches learn from large-scale visuomotor datasets to achieve impressive performance on contact-rich tasks such as shirt folding and whole-body loco-manipulation. However, VLAs inherit the limitations of their training data and design: they lack interpretability

---
[*]Equal contribution  [†]Equal advising [1]NVIDIA [2]UC Berkeley [3]Stanford University [4]Carnegie Mellon University. Correspondence to: Letian Fu <max.fu.letian@berkeley.edu>.

*Proceedings of the 43rd International Conference on Machine Learning*, Seoul, South Korea. PMLR 306, 2026. Copyright 2026 by the author(s).

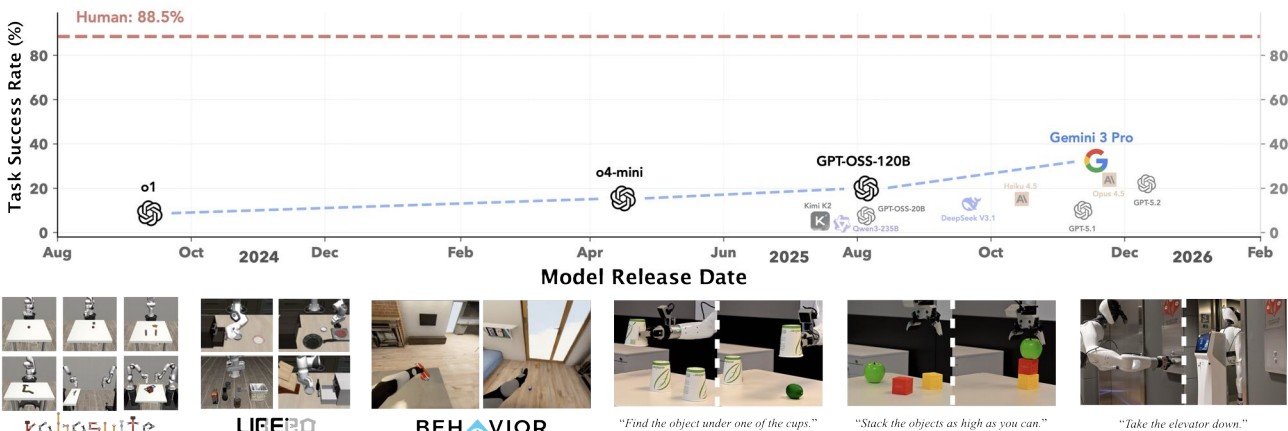

*Figure 1.* (**Top**) CaP-Bench Task Success Rate over Model Release Date: across 7 tasks and 12 models, we compare the success rates of model-generated robot-control programs against those of human expert-written programs. We find that while (vision-) language models have achieved capabilities comparable to humans in other domains (Jimenez et al., 2024; Rein et al., 2024; Hendrycks et al., 2020), they still trail behind human performance in writing code that controls robots for manipulation tasks. (**Bottom**): CaP-Gym integrates Robosuite (Zhu et al., 2020), LIBERO-PRO (Zhou et al., 2025), and BEHAVIOR (Li et al., 2024). We further present CaP-Agent0 (Section 4), a training-free agentic framework that recovers near human-level performance on several manipulation tasks and achieves success rates comparable to—and in some cases exceeding—those of post-trained VLAs, without any task-specific training data.

and struggle to generalize to changes in the environment, new robot embodiments, and long-horizon tasks without additional data collection and retraining.

Recent advances in coding capability of Large Language Models (LLMs) suggest a way to bridge these paradigms: using coding agents to replace the human engineer. Modern coding agents have demonstrated the ability to synthesize executable code, define functions, and debug failures on software engineering benchmarks (Jimenez et al., 2024). Unlike earlier language-conditioned planners limited to function calling (Tellex et al., 2011), today's agents can construct mid- to low-level logic that closely resembles expert code.

Code-as-Policy (CaP) pioneers (Liang et al., 2023; Singh et al., 2023) explored this approach in applications to robotics; they used human-tuned high-level primitives (e.g., `stack_objs_in_order()`) that offer significant task-specific simplifications. As a result, it remains unclear how much of the observed performance in robot control stems from the agent itself versus the structure imposed by these primitives. In particular, prior work does not systematically characterize how agent performance changes as such scaffolding is reduced, nor to what extent increased test-time computation—through iterative debugging, skill synthesis, ensembled reasoning, or multimodal grounding—can compensate for operating over lower-level interfaces.

To address this gap, we introduce **CaP-X**, a unified framework for systematically evaluating and improving code-based robot control agents. At its core is **CaP-Gym**, an interactive environment in which agents directly control robots by generating and executing programs that compose perception and control primitives. CaP-Gym integrates 187 tasks from standard robot manipulation simulators (Robo-

suite (Zhu et al., 2020), LIBERO-PRO (Liu et al., 2023b; Zhou et al., 2025), and BEHAVIOR (Li et al., 2024)) under a shared primitive design that is intentionally compatible with both simulation and physical robot systems.

Building on CaP-Gym, we construct **CaP-Bench**, a benchmark designed to systematically study agentic capability along three axes: **Abstraction Level:** Varying the action space from human-crafted macros (High-Level) to atomic, fundamental primitives (Low-Level); **Temporal Interaction:** Comparing zero-shot single-turn program generation against multi-turn interaction to quantify capabilities in failure recovery and iterative reasoning; and **Perceptual Grounding:** Evaluating how different modalities of visual feedback impact the agent's ability to ground task-relevant visual features into code generation. We instantiate CaP-Bench by focusing on a subset of 7 environments from CaP-Gym and evaluate 12 state-of-the-art open- and closed-source language models and vision-language models.

Guided by insights from CaP-Bench results, we derive **CaP-Agent0**, a training-free agentic framework that augments coding models with multi-turn interaction, visual grounding into text, an automatically synthesized task-agnostic skill library, and parallelized multi-model code generation. CaP-Agent0 achieves performance comparable to, and in some cases exceeding, human expert baselines on CaP-Bench tasks. Finally, we show that CaP-Gym supports **CaP-RL**–reinforcement learning on the coding agent itself. On-policy post-training with environment rewards improves task success, while synthesized programs transfer directly to real robots. This paper makes the following contributions:

1. CaP-Gym, a unified suite of interactive robot coding

*Table 1.* **CaP-Bench evaluation tiers.** Each column specifies a tier by how the agent accesses environment state, primitive abstraction, in-context primitive usage examples, and visual-grounding modality. S1–S4 are single-turn; M1–M4 allows multi-turn interaction.

| | | Single-Turn | | | | Multi-Turn | | | |
|---|---|---|---|---|---|---|---|---|---|
| **Category** | **Characteristic** | **S1** | **S2** | **S3** | **S4** | **M1** | **M2** | **M3** | **M4** |
| Perception | Noiseless (State-Based) | ✓ | | | | | | | |
| | Noisy | | ✓ | ✓ | ✓ | ✓ | ✓ | ✓ | ✓ |
| Primitive Abstraction | High-level | ✓ | ✓ | | | ✓ | ✓ | ✓ | |
| | Low-level | | | ✓ | ✓ | | | | ✓ |
| In-Context Learning | Primitive Usage Examples | | | ✓ | | | | | ✓ |
| Visual-Grounding Modality | Multimodal Feedback | | | | | | ✓ | | |
| | Visual Diff. Module (VDM) | | | | | | | ✓ | ✓ |

environments spanning tabletop, bimanual, and mobile manipulation tasks, designed for evaluating and training code-generating multimodal embodied agents.

2. CaP-Bench, a systematic benchmark that measures robot control performance across tasks and levels of primitives and modalities.

3. CaP-Agent0, a training-free, agentic harness combining multi-turn visual differencing, ensembled reasoning, and automatic skill library synthesis.

4. CaP-RL, reinforcement learning on the coding agent via environment reward.

## 2. CaP-Gym

CaP-Gym is a hierarchical control framework built on the standard **Gymnasium** interface (Brockman et al., 2016). It binds a Low-Level Environment loop (a physics simulator or the real world) with a stateful Code Executor loop. Architecturally, CaP-Gym preserves the native dynamics of underlying simulators, e.g., Robosuite (Zhu et al., 2020), LIBERO-PRO (Liu et al., 2023b; Zhou et al., 2025), and BEHAVIOR (Li et al., 2024), while exposing them through a Read-Eval-Print Loop (REPL) paradigm tailored for coding agents. In CaP-Gym, a code environment "turn" corresponds to one interaction between the coding agent and a specific robot task instance: the agent receives observations, generates a Python program, and the environment executes it to completion. The program may invoke multiple perception and control primitives, each of which can run the simulator or robot controller for multiple internal updates.

### 2.1. Low-Level Perception and Control Primitives

All computationally intensive perception and control primitives are implemented as stateless services (Christie et al., 2025), enabling high-throughput parallel evaluation.

**Perception Primitives** Agents access perceptual data from the environment through modular perception primitives that abstract raw sensor data into structured semantic objects,

e.g., SAM3 (Carion et al., 2025) for language-conditioned segmentation and Molmo 2 (Clark et al., 2026) for open-vocabulary pointing, alongside standard vision libraries like OpenCV (Bradski, 2000) and Open3D (Zhou et al., 2018).

**Control Primitives** Instead of directly emitting joint-space action commands, agents call motion planners or inverse kinematics solvers such as PyRoki (Kim et al., 2025). They can handle collision checking, reachability constraints, and action-space transformations, allowing agents to reason in a task-oriented Cartesian space while delegating execution feasibility to the controller.

## 3. CaP-Bench: Evaluating Frontier Models

**Models.** We evaluate 12 state-of-the-art vision-language and language models, including closed-source frontier models (Gemini-3-Pro (Google DeepMind, 2025), OpenAI GPT o1 (OpenAI, 2024), o4-mini (OpenAI, 2025d), 5.1 (OpenAI, 2025a) and 5.2 (OpenAI, 2025b), and Claude Haiku 4.5 (Anthropic, 2025a) and Opus 4.5 (Anthropic, 2025b), and open source models (OpenAI GPT-OSS-20B and 120B (OpenAI, 2025c), Qwen3 235B (QwenLM / Alibaba Cloud, 2025), Qwen-2.5-Coder-7B-Instruct, Kimi K2 Instruct (AI, 2025), and DeepSeek-V3.1-Terminus (DeepSeek-AI, 2024)).

**Simulation Task Suite.** Primary analysis is performed across 7 core tasks ranging from single-arm manipulation to bimanual coordination: *Cube Lift, Cube Stack, Spill Wipe, Peg Insertion, Cube Re-stack, Two-Arm Lift,* and *Two-Arm Handover*. Each task is evaluated with 100 trials per tier, where each tier specifies the available primitives, interaction mode, and feedback/grounding signal. These 7 tasks are an intentionally controlled core for ablating abstraction, iteration, and grounding under matched conditions; the full release of CaP-Gym ships 187 tasks (7 Robosuite + 130 LIBERO-PRO + 50 BEHAVIOR) for broader community evaluation. In Section 4, we further extend this analysis to diverse long-horizon settings using tasks from LIBERO-PRO (Zhou et al., 2025) and BEHAVIOR (Li et al., 2024).

**Protocol**. We evaluate models using **Zero-Shot**

Task: Lift the red cube

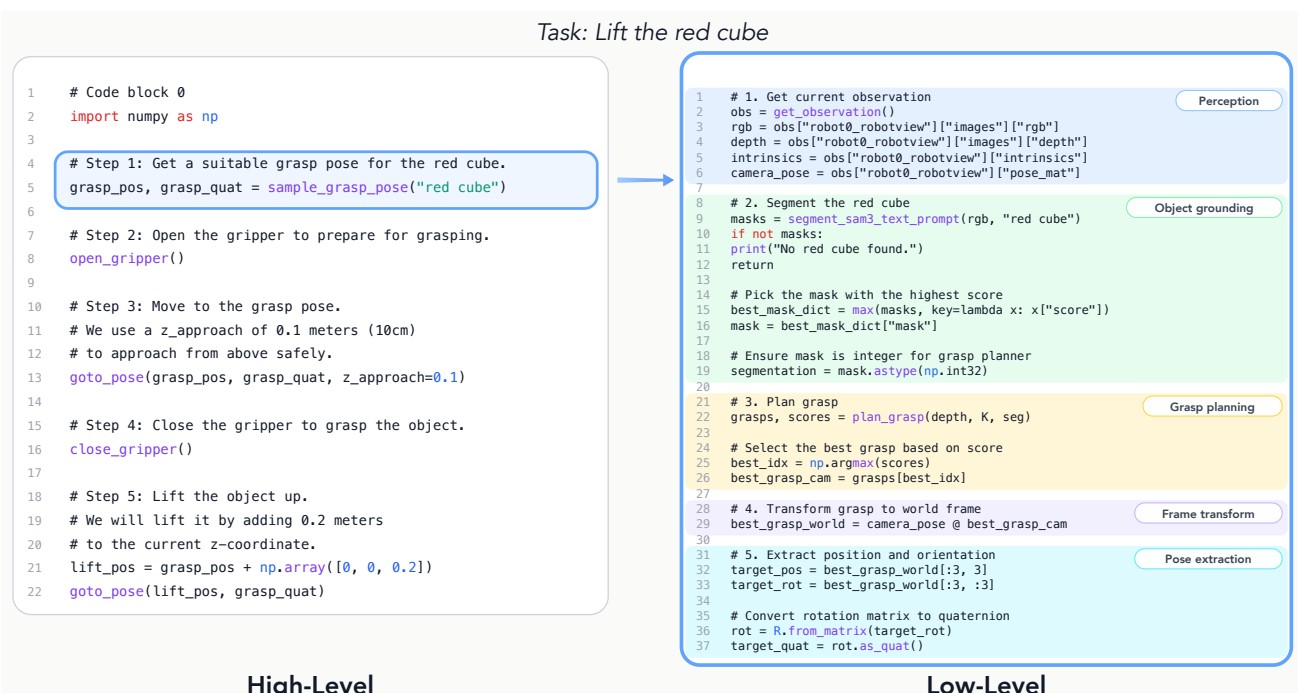

Figure 2. (**Left**) An example of code generated by Gemini-3-Pro for completing the task *"lift the red cube"* using the high-level primitives. (**Right**) Code generated by Gemini-3-Pro using just low-level primitives to achieve the same functionality as a single high-level primitive. For more details on the differences between high and low-level primitives, see Section 3.1 and Appendix G.

**Pass@1** (Chen et al., 2021; Jimenez et al., 2024). Within a trial, agents may interact with the environment over one or multiple turns, observing execution feedback and generating subsequent code to recover from errors or extract additional information; the environment is never reset during the trial. Coding agent task performance is compared against that of human expert-written reference solutions under identical environments and primitives (see Appendix K.1). We introduce 4 single-turn tiers (S1-S4) and 4 multi-turn tiers (M1-M4) to CaP-Bench. Refer to Table 1 for a tabular comparison. The tiers vary three axes faced by any code-as-policy practitioner: (1) *primitive abstraction* (human macros vs. low-level primitives), (2) *level-of-iteration* (single-turn vs. multi-turn with structured execution feedback), and (3) *mode of grounding* (raw visual inputs vs. text descriptions from a separate VLM). CaP-Bench isolates these axes via independently controllable tiers.

### 3.1. Single-Turn Benchmarks (S1-S4)

**High-Level (S1 & S2)**: These tiers evaluate the agent's ability to reason with human-designed primitives. We evaluate this in two modes: **Privileged (S1)**, which uses ground-truth simulation state (masks and object poses), and **Non-Privileged (S2)**, which relies on real perception modules processing raw RGB-D inputs–the default setting for most prior work. We introduce S1 to disentangle high-level planning from perception noise, establishing a reasoning upper bound that allows us to distinguish between algorithmic

failures and visual estimation errors.

**Low-Level (S3 & S4)**: In these tiers, human-designed abstractions are replaced by their constituent low-level primitives (e.g., `solve_ik()`, `sam3_text_prompt()`), drawn directly from the APIs of each underlying package—reflecting the interfaces that human developers use to control robots. We evaluate two settings: **(S3)**, where documentation includes usage examples to scaffold low-level composition, and **(S4)**, where examples are removed and the agent must reason about program structure solely from interface definitions (function signatures and docstrings). See Figure 2 for an illustration of high-level (S1 & S2) versus low-level (S3 & S4) primitives; full API details at each abstraction level are provided in Appendix G.

### 3.2. Multi-turn Benchmarks (M1-M4)

**Text-Only Multi-turn (M1)**: In this setting, the agent receives the standard output (`stdout`) and error traces (`stderr`) from the Python sandbox after each execution turn. This enables a state introspection loop: agents can proactively inject diagnostic print statements to surface hidden symbolic variables (e.g., perception estimates) and utilize these traces to diagnose logical failures and refine code without access to visual ground truth. All other multiturn tiers (M2-M4) retain access to these code execution traces.

**Multimodal (M2)**: The environment pipes the current RGB observation back into the agent's context window. The M2

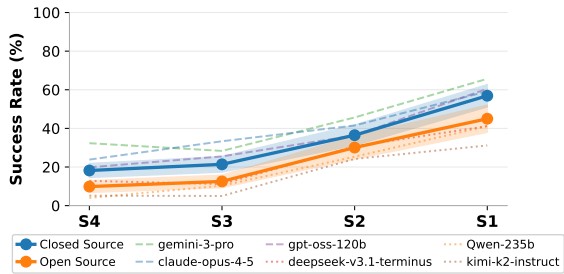

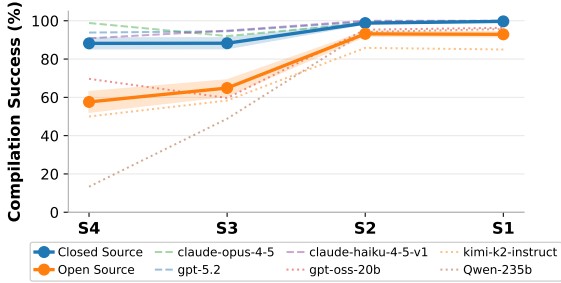

*Figure 3.* Average task success rate across open-source and closed-source models as primitive abstraction increases. As primitive abstraction increases S4 (per model performance illustrated in Figure 1) to S1 (high level primitives with full state observation), the success rate increases. This can only in part be attributed to reduced code correctness at S3-S4, see Figure 4.

tier is only available to multimodal foundation models that accept raw RGB images as input.

**Visual Differencing Module (M3)**: We introduce the Visual Differencing Module (VDM), which uses a vision–language model to convert visual observations into structured natural language. VDM is provided with the task instruction alongside visual observations. In the first turn, it generates a scene description and extracts task-relevant visual attributes. In subsequent turns, it explicitly describes differences between the previous and current image observations and whether the coding agent has completed the task. The resulting text from the VDM is provided as part of the coding agent's observation context for code generation.

**Low-Level with VDM (M4)**: This tier has the same VDM as tier M3 and has access to the same low-level primitives and in-context usage examples as tier S3.

### 3.3. Discussion

**Takeaway 1: A Significant Gap Persists Between Frontier Models and Human Experts in Single-Turn Evaluation.** We benchmark open- and closed-source agents against human expert solutions under an identical set of perception and control primitives in a single-turn, zero-shot setting (S4). The human reference is a near-upper-bound: $N=7$ paper authors (each with 2+ years of robotics-programming experience) wrote single Python scripts at each tier using exactly the same API primitives available to the model, iterating and debugging until the solution achieved $88.5\%$ average single-turn success (see Appendix K.1 for the protocol and effort budget). As shown in Figure 1, while closed-source models consistently outperform open-source alternatives and newer architectures exhibit stronger capabilities, none yet match the success rate of human-crafted programs in a zero-shot Pass@1 setting.

**Takeaway 2: High-Level Abstractions Boost Performance but Limit Expressivity**. Figure 3 shows a monotonic increase in task success as primitive abstraction in-

*Figure 4.* Code Execution Success Rate vs primitive abstraction. creases, mirroring how prior Code-as-Policies (Liang et al., 2023) approaches relying on high-level primitives report strong zero-shot performance. By collapsing low-level perception, geometric reasoning, and control into human-designed primitives, these abstractions reduce the effective search space and allow models to focus on task sequencing.

However, this gain comes at the cost of expressivity. As abstraction increases, the agent's action space is increasingly constrained by human priors, imposing a generality ceiling that masks failures in low-level reasoning. In contrast, performance degradation at lower abstraction levels (S3/S4) reflects the difficulty of code synthesis, while enabling expressive behaviors, such as hierarchical perception fallback strategies (Appendix F.2.2), that cannot be represented by fixed high-level primitives.

This observation motivates a scalable middle ground: rather than relying on human-designed abstractions, agents should be able to recover structure from low-level primitives themselves. In Section 4, we demonstrate this capability by enabling agents to distill successful execution traces into a reusable skill library. Consequently, we propose that generalist embodied coding agents be evaluated primarily on primitive-level performance, ensuring that success stems from robust reasoning rather than from the inductive biases of an over-engineered, often task-specific, API set.

**Takeaway 3: Closing the Loop with Multi-turn and Visual Grounding Improves Performance.** We study how *multi-turn interaction* and different forms of *visual grounding* mitigate the performance gaps identified in Takeaways 1 and 2. Allowing agents to iterate and inspect their own execution traces (`stdout`/`stderr`, M1) consistently improves performance across all models (Figure 5), highlighting the importance of explicit execution feedback for debugging and recovery.

Counter-intuitively, directly interleaving raw RGB observations at each turn (M2) degrades performance relative to the text-only M1 baseline. We hypothesize this degradation is due to a cross-modal alignment gap: foundation models are rarely trained to jointly reason over software coding and images of physical task execution, making raw visual inputs difficult to integrate effectively during code synthesis.

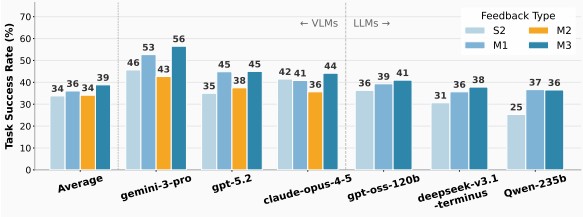

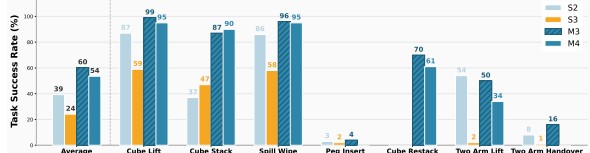

*Figure 5.* Comparison of single-turn (S2) and multi-turn tiers (M1-M3) across models. Enabling textual feedback through multi-turn (M1) improves task success rate in most models. Direct multi-modal (M2) visual grounding reduced task success rate. We find that visual differencing into text (M3) instead of direct multimodal visual input consistently improves task success rate across both close and open source models.

Prior work (Hu et al., 2025; Wang et al., 2026) similarly observes that textually grounded feedback outperforms raw images, though primarily in structured environments with native text states. In contrast, robotic manipulation operates in unstructured, continuous settings without ground-truth textual descriptions. Here, the Visual Differencing Module (M3) bridges this gap by converting visual observations into structured natural language, substantially outperforming both naive image interleaving (M2) and execution-only feedback (M1) across all tasks (Figure 5).

*Low-Level Primitives with Multi-turn Feedback (M4).* Figure 6 shows that agents operating over low-level primitives augmented with multi-turn feedback not only surpass high-level single-turn (S2) but can reach parity with high-level multi-turn performance (M3). This supports a *test-time compute scaling* hypothesis: robustness can be synthesized at runtime by increasing an agent's capacity for reasoning, verification, and self-correction over atomic primitives.

Across all multi-turn settings (M1-M4), both coding errors (e.g., exceptions) and physical execution failures (e.g., unstable grasps) are frequently recoverable through iterative interaction. In the absence of explicit visual grounding mechanisms (M1), successful agents compensate by proactively instrumenting perception primitives to expose symbolic state–such as object poses–and performing explicit checks for task completion (e.g., verifying relative object heights to confirm stacking). Visual grounding (M2-M4) reduces the burden of such self-instrumentation but does not eliminate explicit state verification behavior through perception primitives. Instead, the strongest performance emerges when agents combine structured feedback with iterative reasoning, framing multi-turn interaction as a mechanism for hypothesis testing and error recovery in embodied control.

## 4. CaP-Agent0: An Agentic Framework for Robot Control

Based on the failure modes and insights identified in CaP-Bench, we introduce **CaP-Agent0**, a training-free, agentic

*Figure 6.* Evaluation of Gemini-3-Pro across four tiers of the benchmark. Multi-turn with visual differencing and low-level API (M4) significantly outperforms both single-turn low-level API (S3), and more notably, single-turn high-level API (S2).

infrastructure for robot control that augments base foundation models with a specialized multi-turn reasoning loop and a dynamically synthesized skill library. The architecture of CaP-Agent0 is designed directly to address three key gaps identified in the benchmark. We demonstrate the efficacy of each design choice in CaP-Agent0 by running ablation studies using the most capable model identified from CaP-Bench (Gemini-3-Pro) bootstrapped with each design choice, shown in Figure 8. A visual walkthrough of the agentic framework is presented in Figure 7.

**1. Multi-turn Visual Differencing (VDM):** Directly adopting the insights from Takeaway 3, CaP-Agent0 integrates the Visual Differencing Module as part of the per-turn observation. By grounding observations in structured text rather than raw pixels, the agent mitigates the specific cross-modal alignment failures identified in the M2 tier.

**2. Auto-Synthesized and Persistent Skill Libraries:** In analyzing low-level S3 and S4 code generations from CaP-Bench, we found that capable models routinely synthesize helper functions to perform robotics data manipulations.

Motivated by agentic systems that accumulate reusable tools over time (Wang et al., 2023a), **CaP-Agent0** introduces an *automatically synthesized, task-agnostic skill library* that persists across trials. Rather than requiring the agent to repeatedly re-derive low-level utilities, the library onboards commonly recurring implementation patterns and offloads fragile low-level logic, allowing the coding agent to focus on high-level semantic planning. Importantly, unlike fixed human-designed high-level APIs, these skills are *discovered*: they emerge from successful executions and retain the expressivity of low-level interfaces while improving robustness through reuse.

The library is constructed via an automated synthesis pipeline that can be executed by the agent itself. Specifically, we collect all successful S3-tier rollouts pooled across all 12 models and 7 Robosuite tasks (i.e., the library is not model-specific), extract function definitions via regular-expression matching, and prompt Gemini-3-Pro to identify frequently recurring, task-agnostic logic. This yields a compact library of 9 verified, task-agnostic primitives (Appendix H.1). While the current implementation performs a single synthesis pass, the process is inherently *iterative*. As additional successful executions are accumulated, the agent can con-

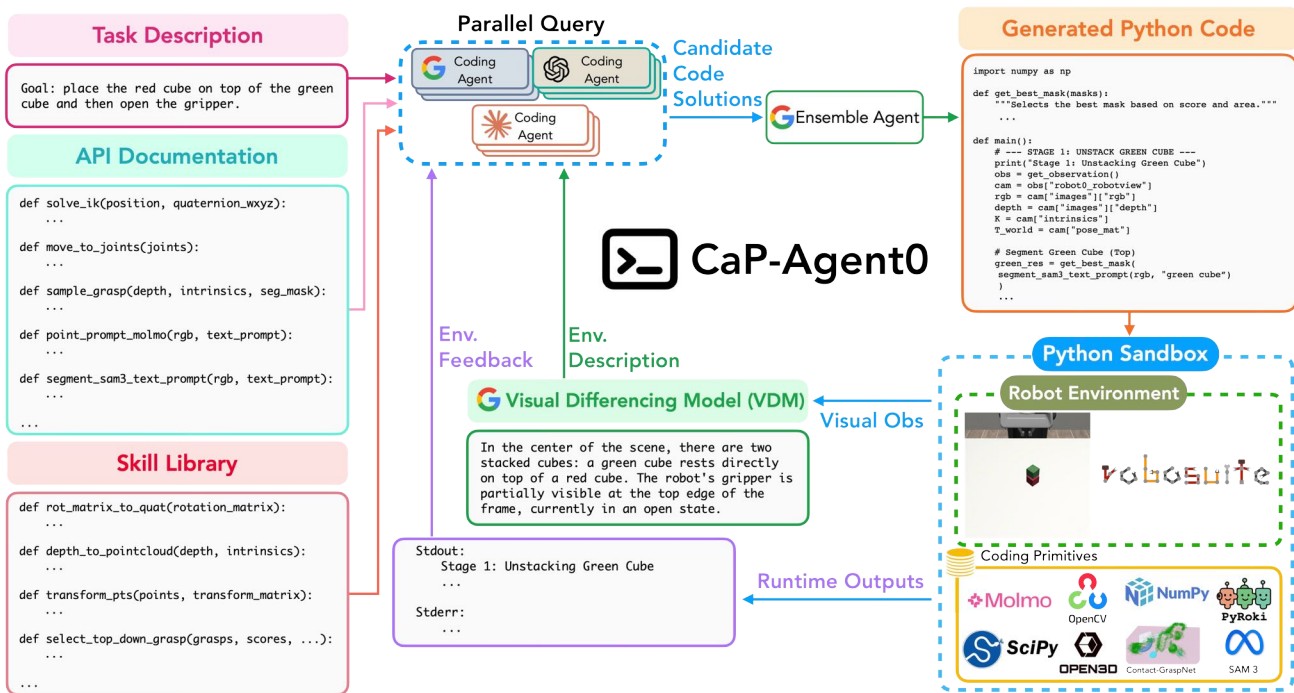

Figure 7. **CaP-Agent0.** CaP-Agent0 incorporates an auto-synthesized skill library of auxiliary utility generated by coding agents during CaP-Bench, a visual differencing model (VDM) which provides a textual description of the initial scene and for each subsequent turn what changes have occurred in the scene since, and a parallel reasoning system where multiple coding agents are provided the same prompt. These coding agents then generate candidate code solutions that may solve the task and the ensemble agent must then synthesize these code generations into a final code snippet which is then executed in the Python sandbox containing the robot environment, whether that environment may be a simulator like Robosuite or on a real robot. For more details see Section 4.

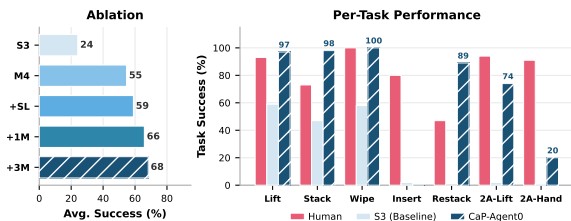

Figure 8. Ablation for CaP-Agent0. (Left): Combining VDM (M4), skill library (+SL), and parallel queries (+1M: Gemini-3-Pro, +3M: Gemini-3-Pro, GPT-5.2, and Claude Opus) significantly improves over single-turn setup with low-level API. (Right): On 4 out of 7 CaP-Bench tasks, CaP-Agent0 achieves comparable or better success rates than human expert code in a single-turn setting.

tinue to update its skill library over time. A complete list of the nine synthesized functions currently utilized by CaP-Agent0 is provided in Appendix H.1.

**3. Parallel Reasoning:** Results from tiers S2 and S3 indicate that failures often stem from insufficient test-time exploration rather than a lack of capability. To address this, CaP-Agent0 employs a parallel reasoning strategy inspired by recent ensemble methods (Pan et al., 2025; Jin et al., 2025; Rodionov et al., 2025). At each turn, the system concurrently samples candidate solutions via two configurations: Single-model: 9 queries to one model. Multi-model: 3 queries each to GPT-5.2, Claude Opus 4.5, and Gemini-3-Pro. To maximize output diversity, we vary sampling

temperatures (see Appendix H.2). A central coding agent then synthesizes these candidates into a final code snippet. This parallel approach is applied to both code generation and to the decision-making process for multi-turn continuations.

### 4.1. CaP-Agent0 Performance on CaP-Bench

Evaluated over 100 trials per task, CaP-Agent0 significantly outperforms single-turn baselines by integrating visual differencing, a self-synthesized skill library, and parallel reasoning (Figure 8). Despite operating solely on low-level primitives, the system achieves success rates comparable to or exceeding human-written programs on 4 out of 7 tasks, narrowing the gap toward expert-level performance.

### 4.2. CaP-Bench++

The 7 core tasks of CaP-Bench isolate abstraction, iteration, and grounding under matched conditions; CaP-Bench++ extends this to compare coding agents directly against (i) state-of-the-art VLA policies and (ii) iterated human-written code on a broader task distribution. Rather than running all 12 models across every extended task–prohibitively expensive at this scale–we focus the comparison on CaP-Agent0. We further evaluate the performance of CaP-Agent0 on subsets of tasks from LIBERO-PRO and BEHAVIOR. We summarize the results of CaP-Agent0 on 30 manipulation tasks

*Table 2.* LIBERO-PRO (Zhou et al., 2025) performance of Open-VLA (Kim et al., 2024), $\pi_0$ (Black et al., 2024), $\pi_{0.5}$ (Physical Intelligence et al., 2025) and CaP-Agent0 on the **libero-object**, **libero-goal**, and **libero-spatial** benchmarks under initial position perturbations (Pos) and instruction perturbations (Task) averaged across tasks. Detailed individual task-wise performance can be found in Appendix J.

| Method | libero-object | | libero-goal | | libero-spatial | |
|---|---|---|---|---|---|---|
| | Pos (Avg.) | Task (Avg.) | Pos (Avg.) | Task (Avg.) | Pos (Avg.) | Task (Avg.) |
| OpenVLA | 0.00 | 0.00 | 0.00 | 0.00 | 0.00 | 0.00 |
| $\pi_0$ | 0.00 | 0.00 | 0.00 | 0.00 | 0.00 | 0.00 |
| $\pi_{0.5}$ | 0.17 | 0.01 | **0.38** | 0.00 | **0.20** | 0.01 |
| CaP-Agent0 | **0.22** | **0.18** | 0.26 | **0.17** | 0.12 | **0.14** |

*Table 3.* Results on BEHAVIOR (Li et al., 2024) Tasks

| Task | Nav. Success Rate | | | Task. Success Rate | | |
|---|---|---|---|---|---|---|
| | Human | S3 | CaP-Agent0 | Human | S3 | CaP-Agent0 |
| Pick up Radio | **88%** | 72% | 80% | 36% | 24% | **56%** |
| Pick up Soda Can | 80% | 52% | **84%** | **72%** | 32% | **72%** |

from LIBERO-PRO in Table 2 and compare them against three state-of-the-art VLA methods: OpenVLA (Kim et al., 2024), $\pi_0$ (Black et al., 2024), and $\pi_{0.5}$ (Physical Intelligence et al., 2025). Although these VLA methods are training-based, CaP-Agent0 is a training-free method that has comparable or exceeds the performance of VLA post-training on these tasks. LIBERO-PRO (Zhou et al., 2025) extends the LIBERO (Liu et al., 2023b) benchmark by increasing the perturbations of tasks by reasonable amounts along object attribute perturbations, initial position perturbations, instruction perturbations, and environmental perturbations. CaP-Agent0 and these VLA methods are evaluated along initial position perturbations (Pos) and instruction perturbations (Task). Under Pos perturbations, the initial positions of objects in the scene are swapped with one another (ex: the position of the frypan and moka pot) and under Task perturbations, the instruction is changed so that another object in the scene is manipulated (ex: "Put the moka pot on the stove" → "Put the frypan on the stove"). Since VLAs are trained on a different instruction distribution, they perform poorly under task perturbations, whereas CaP-Agent0 remains robust to instruction variations. Furthermore, CaP-Agent0 achieves performance comparable to $\pi_{0.5}$ under initial position perturbations.

We evaluate our method on two long-horizon mobile manipulation tasks in BEHAVIOR, where an R1Pro wheel-based humanoid is required to pick up a radio from a table and a soda can from the floor. The results over 25 trials for each task are summarized in Table 3, reporting both navigation success rates and task completion success rates. In both tasks, the robot may start with the target object outside its field of view and must actively search for and navigate toward it. Navigation is considered successful if the robot reaches a location within 1 m of the target.

The robot can adjust its camera in both horizontal and ver-

tical directions and move both its base and arm to reach the object, resulting in a substantially larger action space than in tabletop manipulation settings. Moreover, potential collisions with surrounding furniture may prevent the robot from reaching the desired pose, leading to partial execution of planned trajectories and increased task complexity.

In the radio pickup task, although the robot often successfully locates and approaches the object, it may lose sight of it or encounter severe occlusions when navigating too closely, due to its limited field of view. This frequently results in missing or poor grasp poses and constitutes a major failure mode for both S3 and the human policy. In contrast, CaP-Agent0 mitigates this issue by repositioning the robot to obtain a better view, achieving substantially higher success rates. For the soda can pickup task, the small object size makes accurate vertical camera alignment critical for localization. S3 often fails by adjusting the camera vertically too early, leading to unsuccessful searches within the time limit. CaP-Agent0 adapts its search strategy based on feedback, improving navigation success. Furthermore, grasping may fail when the can is knocked over, further reducing S3's completion rate. CaP-Agent0 can resample grasp poses after such disturbances and successfully recover, achieving significantly higher task success and similar performance as human expert.

### 4.3. Let the Agent Experience the Real World

The design of the CaP-Gym environment loop deliberately allows it to directly interface with real-world robot perception and control interfaces, and we additionally demonstrate zero-shot performance of CaP-Agent0 in unseen real-world tasks on real robot embodiments including the Franka Panda and AgiBot G1 without requiring any major cross-embodiment modifications (with the exception of single arm to bimanual control primitive modifications). With no post-training, off-the-shelf VLMs such as Gemini-3-Pro and Claude Opus 4.5 can perform complex long-horizon robotics reasoning and manipulation tasks following natural language instructions. For example, when asked to "find the object under one of the cups" (Figure 12), it mechanically searches through all cups with closed-loop feedback from vision. When the robot is asked to solve a math problem presented in the physical world (Figure 13), it perceives the visual cue, thinks, and selects the correct blocks on the first attempt. We also demonstrate embodied reasoning ability in Figure 15, where CaP-Agent0 exhibits common sense physics reasoning and understands the sensible stacking order for objects of various shapes. Optionally, a human-in-the-loop can also interactively correct the robot's behavior by providing additional feedback in between turns. More details are documented in Appendix B.

*Table 4.* **Impact of RL Post-Training in Sim and Real.** Comparison of success rates between the base model, our RL post-trained agent, and human experts. Simulation results are averaged over 100 trials per task, while real-world deployment on a Franka Emika robot is evaluated over 25 trials.

| | Simulation (N=100) | | | Real World (N=25) | |
|---|---|---|---|---|---|
| Method | Cube Lift | Cube Stack | Spill Wipe | Cube Lift | Cube Stack |
| Human Expert | 93% | 73% | 100% | 92% | 84% |
| Qwen 2.5 Coder 7B | 25% | 4% | 30% | 24% | 12% |
| Qwen w/ CaP-RL | **80%** | **44%** | **93%** | **84%** | **76%** |

## 5. CaP-RL

CaP-Gym enables on-policy reinforcement learning with verifiable rewards (RLVR) directly on the coding agent. To demonstrate this, we apply Group Relative Policy Optimization (GRPO) (Shao et al., 2024; Guo et al., 2025) to post-train a Qwen2.5-Coder-7B-Instruct base model (Hui et al., 2024).

**Methodology.** We RL post-train on three tasks: *Cube Lift, Cube Stack,* and *Spill Wipe*. To ensure stable convergence, we train using the privileged state-based APIs of tier S1. This avoids the noisy reward signals present in tier S2, where compounding perception and control errors can cause otherwise correct programs to fail during execution, introducing credit assignment ambiguity similar to that observed in G1 (Chen et al., 2025).

**Simulation Results.** Post-training for 50 iterations per task significantly improves code compilation rates and strategic robustness. When evaluated on S2 (noisy perception), the RL post-trained model achieves substantial gains over the base model, as detailed in Table 4.

**Sim-to-Real Transfer.** A key property of CaP-RL is that what transfers across the sim-to-real boundary is the *code-as-action-space*: the agent learns to compose shared perception and control tools that are fixed across simulation and reality, rather than mapping raw visual features to motor commands. We validate this on a Franka Emika robot, where the agent retains high success rates for cube lifting (84%) and stacking (76%), demonstrating that strategies learned in simulation remain robust under real-world perception noise on these tasks. Please refer to Appendix E for comparing code generation before and after RL post-training.

## 6. Related Work

**Code as Policies.** A growing line of work explores programmatic robot control, where LLMs generate executable code that orchestrates perception and control modules (Gupta & Kembhavi, 2023; Yao et al., 2022). In robotics, this paradigm spans grounding plans in affordances (Ahn et al., 2022), composing APIs for closed-loop behaviors (Singh et al., 2023; Liang et al., 2023), generating Python programs over perception APIs (Huang et al., 2023; Mu et al., 2024; Goldberg et al., 2025), and modular vision-language agen-

tic pipelines (Shi et al., 2025; Huang et al., 2022b; Team et al., 2025). Structured intermediate representations such as PDDL (Aeronautiques et al., 1998) and Signal Temporal Logic (Liu et al., 2023a; Chen et al., 2024) and persistent state tracking (Yoneda et al., 2024) improve plan reliability, while executable code is empirically a superior agent action representation (Wang et al., 2024a), with agents benefiting from iterative self-refinement (Shinn et al., 2023; Madaan et al., 2023) and inference-time sampling (Wang et al., 2023b; Snell et al., 2024). Despite this progress, most coding-agent work on robot control still relies on high-level, human-crafted APIs encoding significant task structure. CaP-Gym exposes the full primitive stack down to joint-level perception and control, and CaP-Bench evaluates frontier models across abstraction tiers and inference-time strategies.

**Skill Synthesis and RL with LLM-Generated Code.** A large body of work uses LLMs as *static* code generators for reward functions, curricula, and skill synthesis with a frozen LLM and a separate trained policy (Ma et al., 2024a; Yu et al., 2023; Liang et al., 2024; Ma et al., 2024b; Du et al., 2023; Wang et al., 2024b; Ahn et al., 2024); complementarily, RL with verifiable rewards (RLVR) improves the model itself across reasoning, code, and agentic settings (Guo et al., 2025; Shao et al., 2024; Wei et al., 2025; Pan et al., 2024; Feng et al., 2026). CaP-RL extends RLVR to robot manipulation by directly fine-tuning the language model via GRPO on physics-simulation outcomes, rather than using the LLM to produce reward code for a separate policy.

**Benchmarks for robotics and embodied agents.** Robotic manipulation benchmarks (Zhu et al., 2020; Liu et al., 2023b; Li et al., 2024; Mees et al., 2022; James et al., 2019; Yu et al., 2020) evaluate fixed policy interfaces rather than executable program synthesis with tiered APIs and multi-turn debugging; code benchmarks (Chen et al., 2021; Jimenez et al., 2024; Pan et al., 2024) lack embodied perception; and embodied-agent benchmarks (Yang et al., 2025; Chen et al., 2025; Wang et al., 2026; Liu et al., 2024; Shridhar et al., 2020) broaden to multimodal interactive environments but do not require executable robot-control code across abstraction tiers. CaP-Gym targets this intersection.

## 7. Conclusion

We introduce CaP-X, a unified framework for benchmarking and improving coding agents for robot control. CaP-X consists of CaP-Gym, CaP-Bench, CaP-Agent0, and CaP-RL, enabling controlled evaluation across abstraction levels, interaction modes, and learning paradigms. CaP-X frames robot control as a problem of machine intelligence—where agent design, inference-time computation, perception, and control are co-studied—providing a testbed for evaluating and advancing general-purpose embodied intelligence.

## Impact Statement

This paper presents work whose goal is to advance the field of machine learning. There are many potential societal consequences of our work, none of which we feel must be specifically highlighted here.

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

## A. Future Works

Programmatic control performs well on long-horizon, reasoning-heavy tasks, but remains brittle for contact-rich behaviors that require tight visual servoing and continuous feedback (e.g., insertion or pouring). One promising direction is hybrid CaP-VLA policies, in which a coding agent manages high-level task logic and recovery while deferring low-level execution to VLA policies. Results from CaP-Bench further highlight several avenues for improving language-model-based agents, including stronger embodiment-aware planning and reasoning, more effective grounding of task-relevant visual information into code generation, improved test-time search, and agent or prompt optimization methods (Khattab et al., 2024). From a robotics perspective, robustness may further improve by incorporating optimization-based control primitives that allow agents to specify task-level constraints and account for collision avoidance during motion planning, rather than relying solely on inverse kinematics solutions that may be suboptimal when directly interpolated to in joint space. Expanding CaP-Gym to additional environments and classical robotics problems (e.g., active and interactive perception) would further stress-test agentic reasoning.

## B. Interactive Real-World Setup

In this section, we document the interactive real-world experience with CaP-Agent0 centered around a chat-based web UI, where the user can propose new tasks to the robot, view the step-by-step tool-use results, and offer feedbacks for multi-turn improvement. We will show how this system enables a real-world robot (AgiBot G1) to zero-shot complete novel tasks.

### B.1. User Interface

On the chat UI shown in Figure 9, the user start with choosing task configurations and models. Then they can view the scene description and code generated by the model. As the code is executed on the robot, they also see the step-by-step progress of each tool call, such as the segmentation masks returned by SAM3, Points annotated by Molmo 2, and wrist camera views when the robot arm moves. In between turns, the user can provide additional feedback to the model by typing in the chatbox on the bottom.

The UI also features a 3D visualization powered by Viser (Yi et al., 2025), which shows the live robot URDF, trajectories, and depth pointcloud.

### B.2. Tools

CaP-Agent0, specifically on the AgiBot G1, has access to these APIs for perceiving, reasoning, and interacting.

- `get_observation`: returns a dictionary of the current head camera image (RGB, depth, and intrinsics), left and right wrist camera images, end-effector 6-DoF force-wrenches, and joint configuration.

- `segment_sam3_text_prompt`: given an image and a text prompt, returns a list of segmentation masks and scores.

- `segment_sam3_point_prompt`: given an image and a point coordinate on the image, returns a list of segmentation masks and scores.

- `point_prompt_molmo`: given an image and a text prompt, returns a list of points on the image pointed out by Molmo 2.

- `open_gripper`: opens gripper.

- `close_gripper`: closes gripper.

- `goto_pose`: Plans and executes IK to a pose specified by a position and orientation, optionally with a pre-grasp offset and max force-wrench.

- `matrix_to_pose_wxyz_xyz`: Converts a $4 \times 4$ transformation matrix to a pose represented in a quaternion and a 3D position.

- `pose_wxyz_xyz_to_matrix`: Converts a pose to a transformation matrix.

- `euler_to_quaternion_wxyz`: Converts extrinsic XYZ Euler angles to a quaternion.

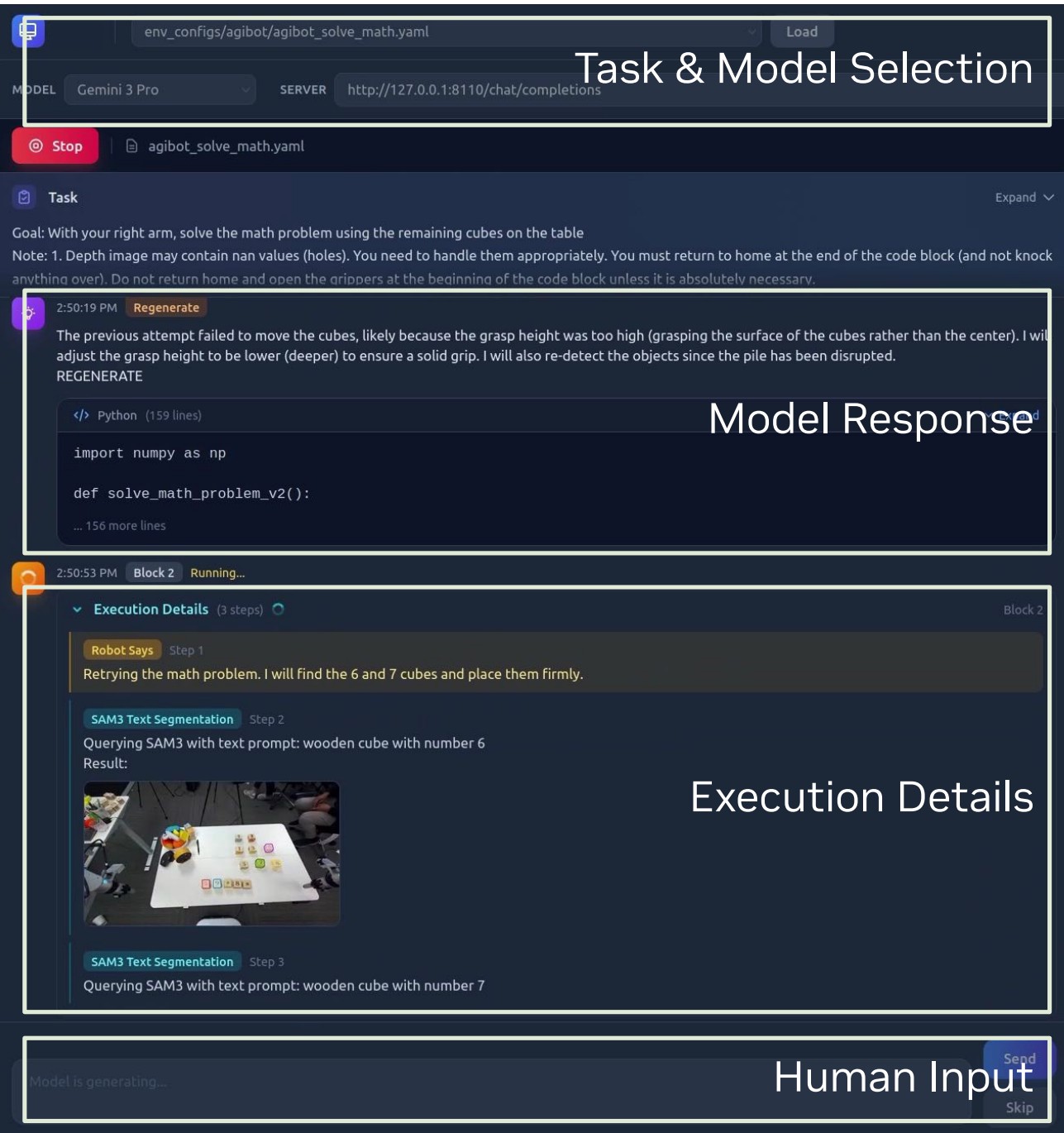

*Figure 9.* An example of the web UI, where users can select the task and model at the top, view the model's responses and execution details in the middle, and provide instructions at the bottom.

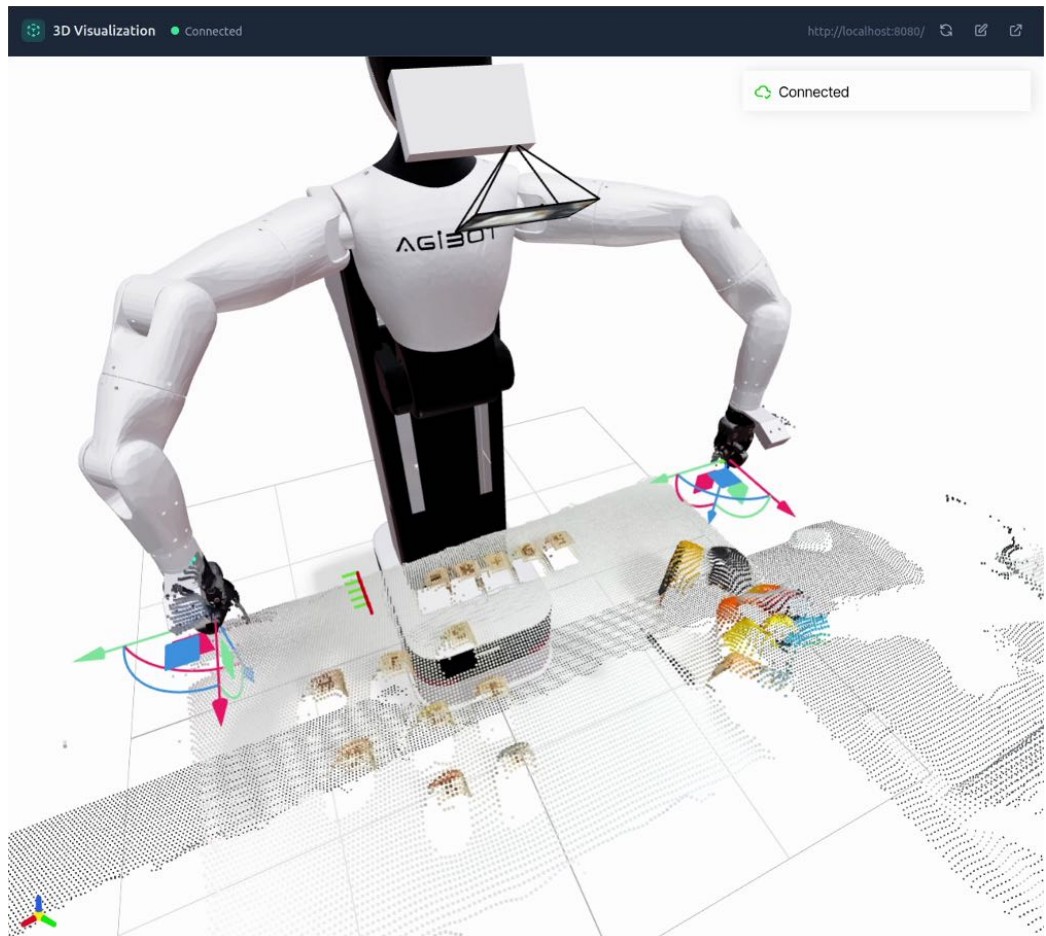

*Figure 10.* Viser (Yi et al., 2025) visualization section of the web UI.

- `quaternion_wxyz_to_euler`: Converts quaternion of extrinsic XYZ Euler angles.

- `query_vlm`: Ask a question with text and images to an expert VLM model (Gemini-3-Pro)

- `go_forward`: Move forward 1 meter.

- `turn_left_45_degrees`: Turn left 45 degrees.

- `turn_right_45_degrees`: Turn right 45 degrees.

- `goto_planar_position`: Plan and execute IK to a position with planar motion constraint, but no orientation (yaw) requirement.

- `convert_depth_to_pointcloud`: Given a depth image, intrinsics, and extrinsics, convert the depth image into a pointcloud.

- `forward_kinematics`: Given a robot configuration, compute the end-effector pose.

- `overlay_eef_axis_on_image`: Visualize end-effector pose on a camera image by rendering the 3D axis on top.

- `overlay_segmentation_masks`: Visualize segmentation masks on an image.

- `say_something`: Convey some intent to the user.

Among the tools, the 3D rigid body transformation helpers are provided although technically not strictly necessary. We find that when not provided, these functions are almost always written by the models who make mistakes from time to time.

Visualization tools such as `overlay_eef_axis_on_image` and `overlay_segmentation_masks` enables advanced multi-modal reasoning when the output images are passed to an expert VLM via `query_vlm`. This allows, for example, code execution conditioning based on if the expert VLM answered "which of these segmentation masks should I pick?".

### B.3. Real World Tasks

In this section, we show visualizations of CaP-Agent0 completing tasks in the real world. All experiments are *zero-shot*. The agent's execution details are condensed and presented as pseudo sub-steps, while CaP-Agent0 writes the underlying code in greater detail using tools available in B.2. Raw code example is available in Appendix F.

### B.3.1. NEEDLE IN A HAYSTACK

In this task, we evaluate the ability of CaP-Agent0 to find "needles in a haystack." Specifically, in a cluttered scene containing diverse objects, the robot is tasked with locating and grasping an auto pencil refill holder, a relatively uncommon item. Such uncommon objects are often challenging for end-to-end learning policies such as VLAs. In contrast, CaP-Agent0 leverages pretrained VLMs to successfully localize and retrieve the target object, as shown in Figure 11.

### B.3.2. MECHANICAL SEARCH

In this task, three inverted cups are placed on a table, and the robot is instructed to retrieve a green lime hidden beneath one of them. This occluded object retrieval problem is commonly referred to as mechanical search (Huang et al., 2022a). Because the lime is concealed by the cups, the robot must systematically explore each cup to locate it. A representative planning and execution example is shown in Figure 12.

### B.3.3. MULTIMODAL SYMBOLIC REASONING

In this task, we evaluate the multimodal symbolic reasoning capabilities of CaP-Agent0 by requiring it to solve a mathematical equation formed by numbered wooden blocks. The robot must first perceive the scene to interpret the equation and then grasp the correct block and place it in correct location to complete the solution, as shown in Figure 13.

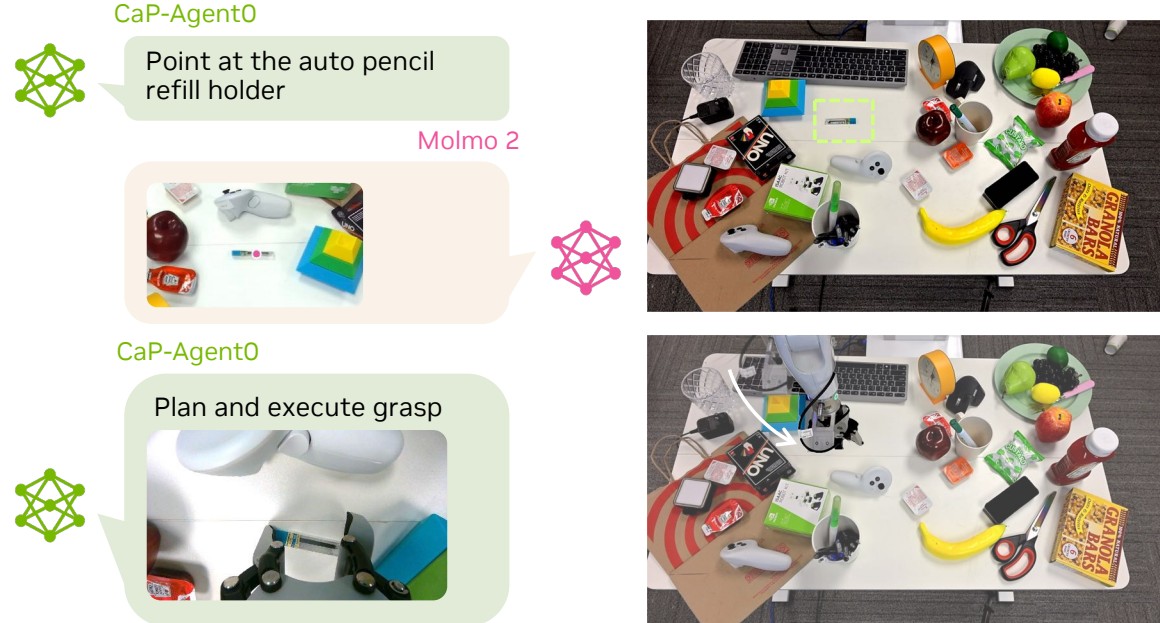

*Figure 11.* An example of CaP-Agent0 locating and grasping an auto pencil refill holder, where it uses Molmo2 to locate the object.

### B.3.4. LEARNING FROM HUMAN FEEDBACK

CaP-Agent0 is able to incorporate human feedback and adjust its code generation accordingly to complete the task. As shown in Figure 14, the robot is instructed to pick up an apple but fails in the initial attempt due to an excessively high grasp pose. After receiving the feedback "grasped the apple too high," CaP-Agent0 modifies the generated code and successfully completes the task on the second attempt.

### B.3.5. EMBODIED REASONING

This task evaluates the embodied reasoning ability of CaP-Agent0, where the robot is asked to "stack objects as high as possible". Because the scene contains both square and round objects, the robot must reason about their physical properties to determine a stable stacking strategy. As shown in Figure 15, CaP-Agent0 learns to place round objects on top of square ones to maximize stability.

### B.3.6. TOOL GENERALIZATION WITH DOMAIN KNOWLEDGE

This task demonstrates the versatility of the code-as-policy interface, where CaP-Agent0 can use tools from arbituary Python packages to assist with the task. We ask the robot to "take the elevator downstairs", which suggests locating and pressing the button, and moving into the elevator. Since the robot is positioned at an angle with respect to the wall, it is not immediately obvious in which direction the button should be pushed. CaP-Agent0 is able to invoke SciPy RANSAC algorithm on the segmented wall pointcloud, and compute the surface normal direction, shown in Figure 16.

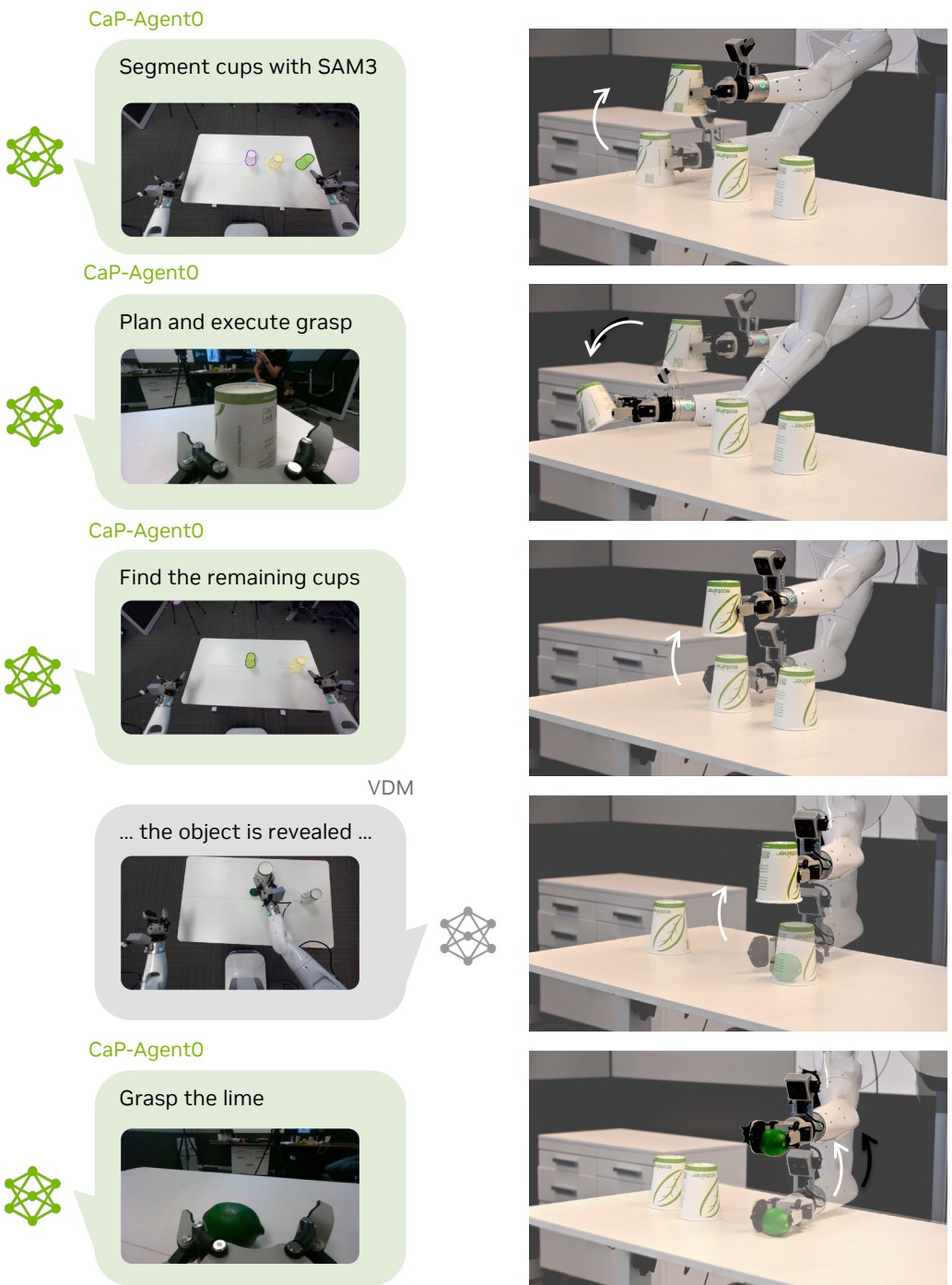

*Figure 12.* Mechanical search planning and execution example. The left column shows the agent planning process and the right column shows the physical robot execution.

Figure 13. Example of a Multimodal Symbolic Reasoning task, where the robot is asked to solve the problem of 59 plus 8.

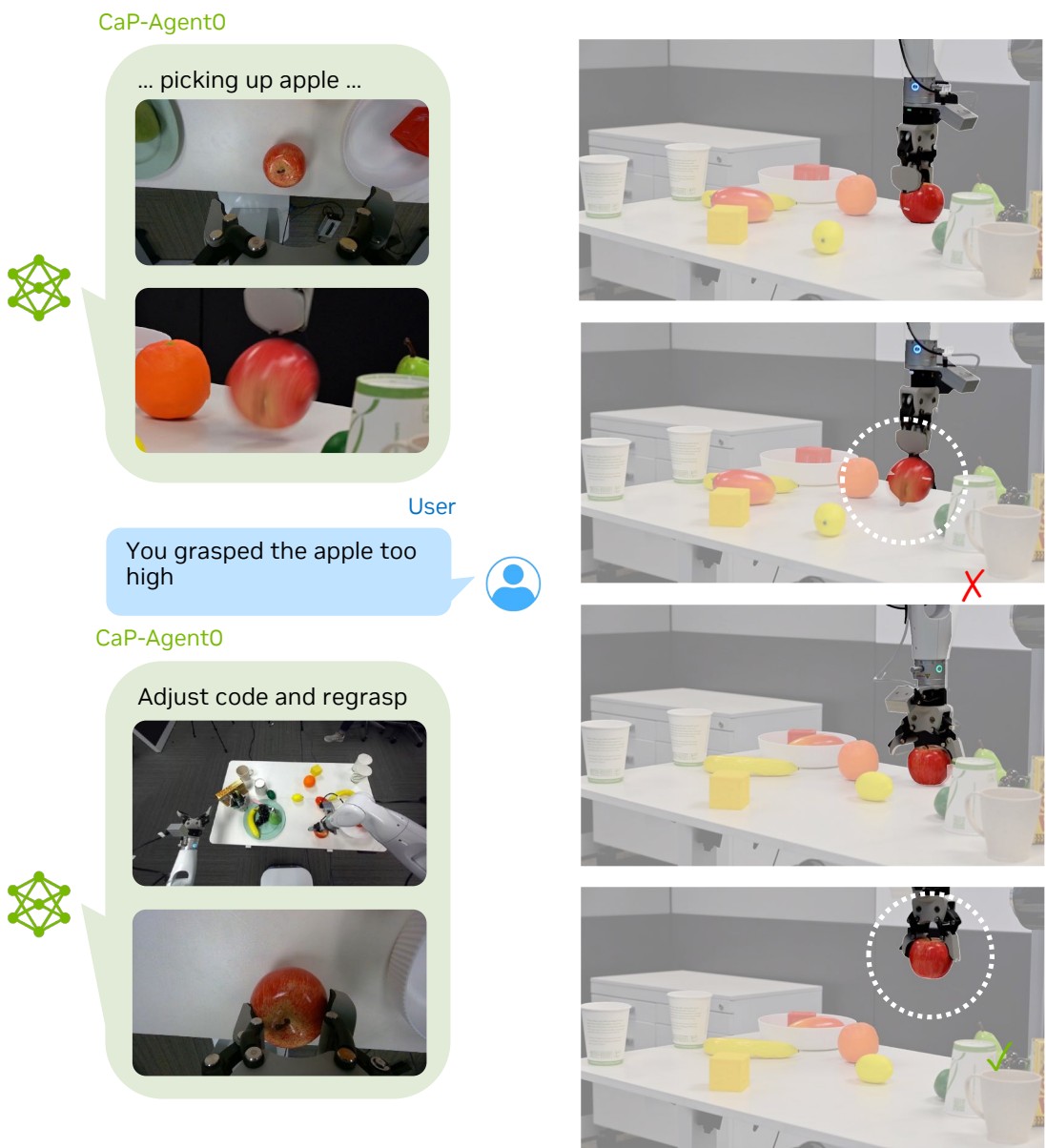

*Figure 14.* Example of picking up an apple. While the first attempt fails, CaP-Agent0 takes human feedback as input, adjusts the code and completes the task successfully.

CaP-Agent0

The apple (round) needs to be stacked on the cubes (square).

CaP-Agent0

Segment with SAM3 and compute bounding box

CaP-Agent0

Stack red cube

CaP-Agent0

Stack apple

*Figure 15.* Example of CaP-Agent0 stacking round objects on top of square objects to form the tallest possible stable stack.

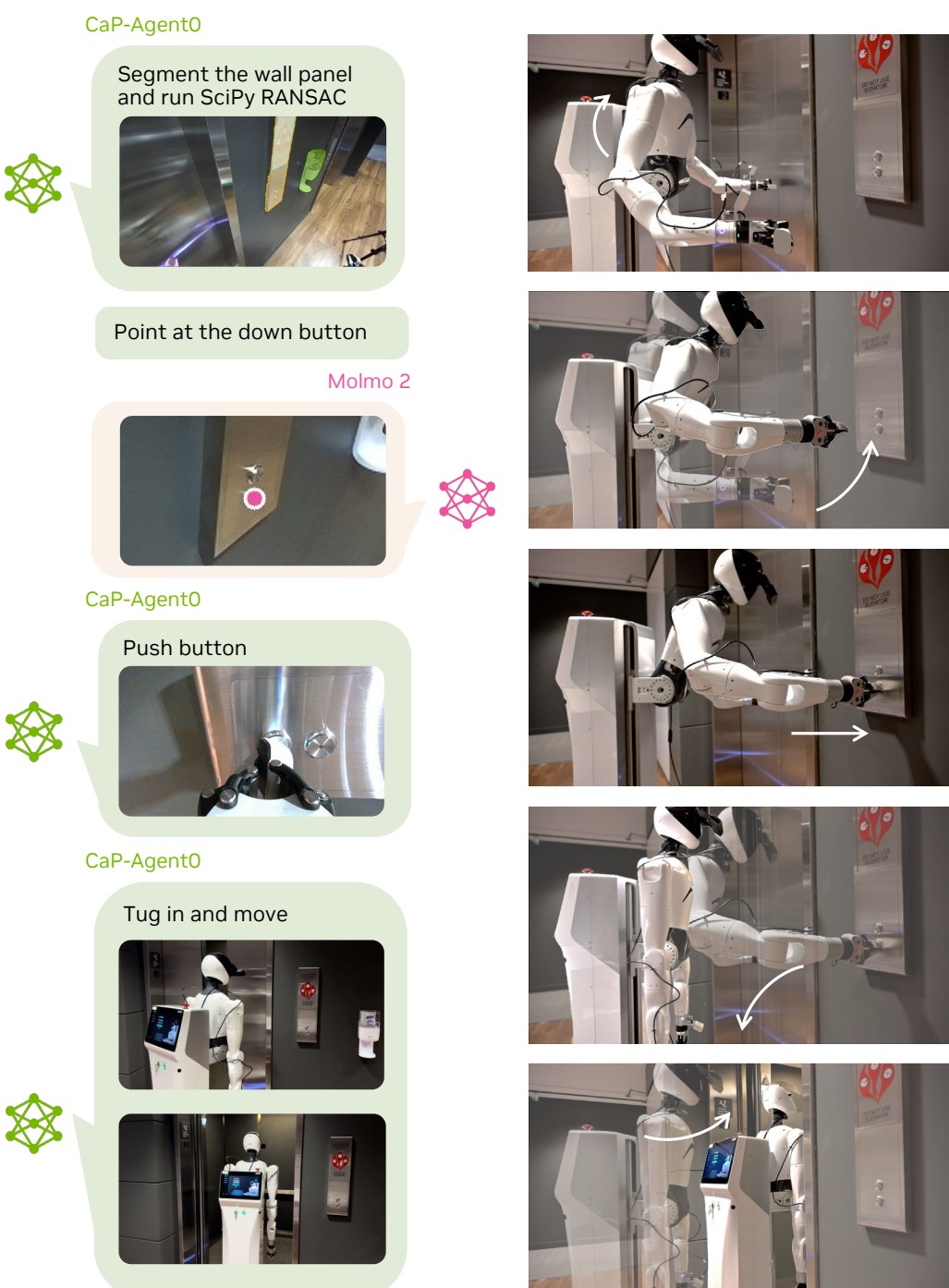

*Figure 16.* Example of robot pushing the elevator button.

# C. Full Benchmark Table

We present the full results from CaP-Bench in Figure 17. From top to bottom are code compilation success rate, average dense reward, and average task success rate.

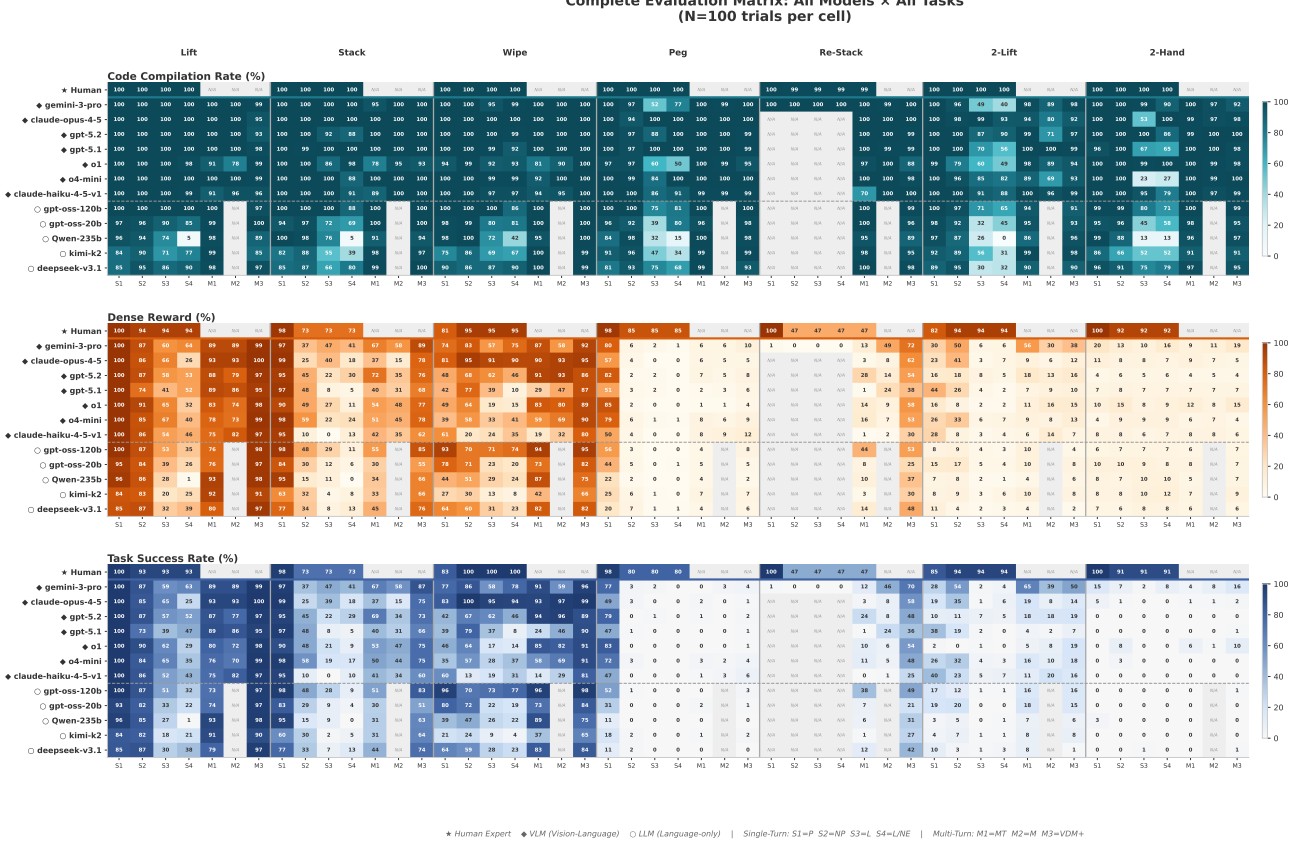

Figure 17. Full CaP-Bench results. From top to bottom are code compilation success rates, average task reward, and average task success rate. Note that for Re-Stack, with a simple task prompt, without visual input, we do not observe model writing code to first check the initial condition and then perform the task, resulting in failures. Therefore, we only evaluatd all models for the multi-turn setup.

# D. Additional Takeaways

**In-Context Examples of API Usage improves performance**. In the docstring of each API used in S4, we provided a brief description of the API, the expected datatype, and shape for each input and output. However, due to low code compilation successes from open source models in the initial experiments, we hypothesize that additional prompting may be necessary to increase model performance. Therefore, in addition to the components above, we added API usage examples in the docstring.

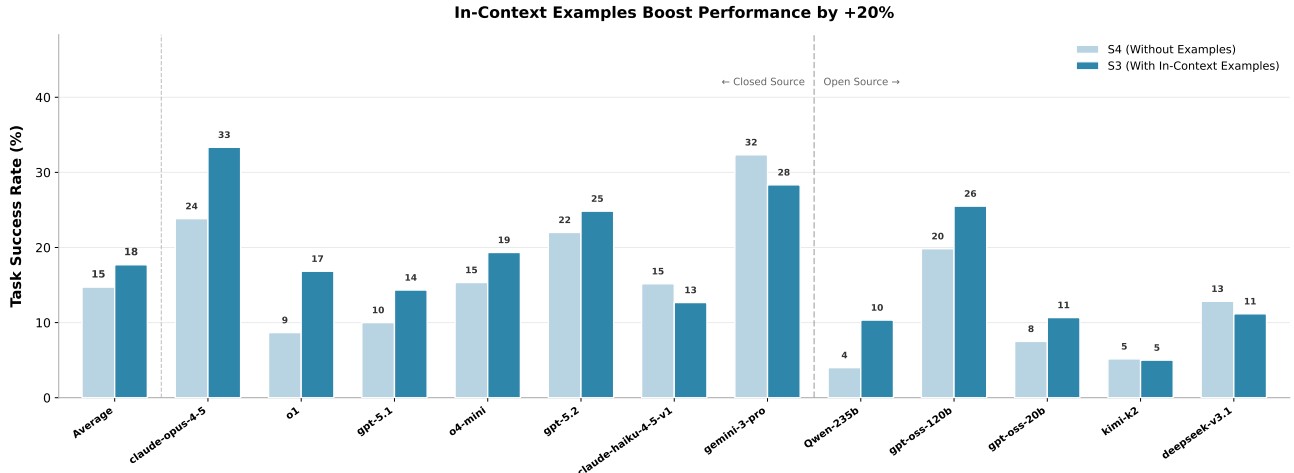

*Figure 18.* Per-model breakdown of usefulness of in-context API Usage

We present the results in Figure 3 and Figure 4, and a per-model breakdown in Figure 18. We observe that almost all models benefited from adding additional context of examples of how to use the low-level APIs.

# E. Qualitative Analysis of CaP-RL Post-Training Effects

In this section, we analyze the qualitative behavioral shifts induced by on-policy reinforcement learning (GRPO) on the `Qwen2.5-Coder-7B-Instruct` base model. We focus on the *Stack Red Cube on Green Cube* task to illustrate how RL mitigates fundamental instruction-following failures and induces robust geometric reasoning. Methodologically, CaP-RL leverages the privileged S1 tier to stabilize training by removing perception noise. Below we show the code block examples before and after CaP-RL Post-Training.

**Before CaP-RL Post-Training Model Behavior.** Qwen2.5-Coder-7B-Instruct base model code generation example on the task *Stack Red Cube on Green Cube* prior to on-policy reinforcement learning with CaP-RL:

```python
import numpy as np

# Get the pose of the green cube
green_cube_position, green_cube_quaternion_wxyz, _ = get_object_pose("green cube")

# Get the pose of the red cube
red_cube_position, red_cube_quaternion_wxyz, _ = get_object_pose("red cube")

# Calculate the placement position for the red cube
placement_position = np.array(green_cube_position) + np.array([0, 0, 0.05])  # 5 cm
    above the green cube

# Move the robot to the placement position
goto_pose(placement_position, (0, 0, 1, 0), z_approach=0.1)  # Use a slight z_approach
    for precision

# Open the gripper to release the red cube
open_gripper()
```

**After CaP-RL Post-Training Model Behavior.** Code generation example of Qwen2.5-Coder-7B-Instruct model after on-policy reinforcement learning with CaP-RL on the task *Stack Red Cube on Green Cube* :

```python
import numpy as np

# Get the positions and extents of the red and green cubes
red_cube_pos, _, red_cube_extent = get_object_pose("red cube", return_bbox_extent=True)
green_cube_pos, _, green_cube_extent = get_object_pose("green cube", return_bbox_extent
    =True)

# Calculate the placement position on top of the green cube
placement_pos = green_cube_pos + np.array([0, 0, red_cube_extent[2] / 2 +
    green_cube_extent[2] / 2])

# Sample a grasp pose for the red cube
grasp_pos, grasp_quat = sample_grasp_pose("red cube")

# Move to the grasp pose and approach
goto_pose(grasp_pos, grasp_quat, z_approach=0.1)

# Close the gripper to grasp the red cube
close_gripper()

# Move to the placement position
goto_pose(placement_pos, (0, 0, 1, 0), z_approach=0.1)

# Open the gripper to release the red cube
open_gripper()
```

**Pre-RL Failure Modes: Step Skipping and Hallucinated State.** Prior to RL post-training, the base model frequently exhibits "step skipping", a failure mode where the agent attempts to satisfy the final goal state without executing the necessary prerequisites. As shown in the pre-RL snippet, the model correctly identifies the target location (`placement_position`) but fails to grasp the red cube. It attempts to move the gripper directly to the placement target and open it, seemingly hallucinating that it is already holding the object.

**Post-RL Improvement: Causal Sequencing and Geometric Generalization.** After GRPO training, two critical improvements emerge:

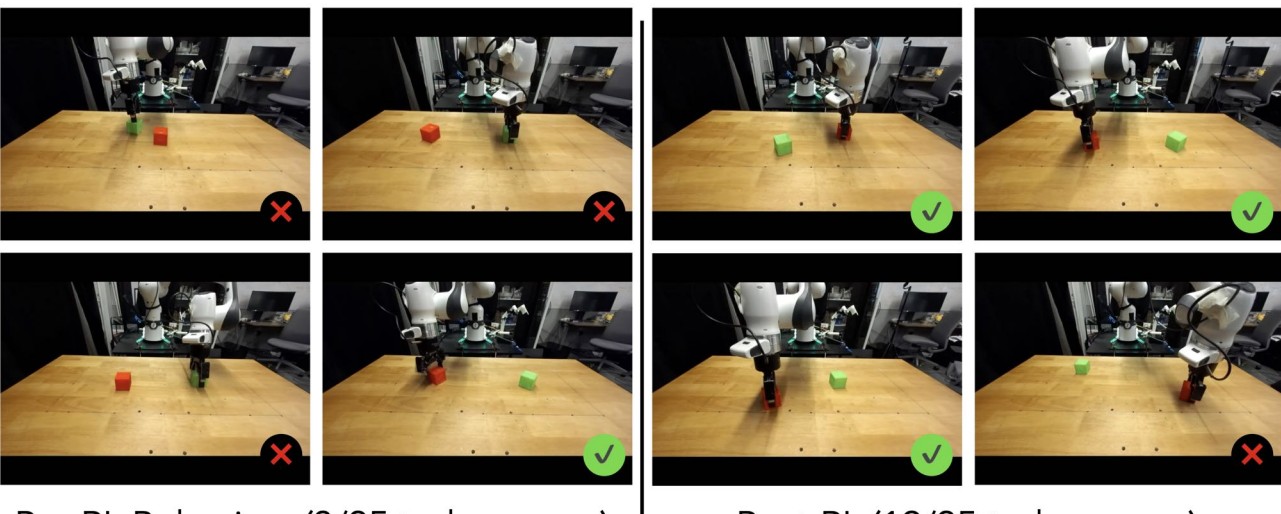

Pre-RL Behaviors (3/25 task success) | Post-RL (19/25 task success)

*Figure 19.* CaP-RL Finetuned Qwen-2.5-Coder-7B-Instruct deployed on the real task *put the red cube on the green cube*. Human Oracle code gets 21/25 task successes in real. Prior to reinforcement learning, we see behaviors such as grasp approaches on the green cube as the first action, rather than a more sensible approach on the red cube first.

1. **Causal Sequencing:** The model correctly synthesizes the full manipulation chain: *Identify → Grasp → Transport → Release*. It learns through environment interaction and reward feedback signals that the causal dependency that an object must be grasped (via `close_gripper`) before it can be placed.

2. **Dynamic Geometric Reasoning:** Instead of using hard-coded offsets, the RL-trained model utilizes the `return_bbox_extent=True` parameter to dynamically calculate the stacking height based on object dimensions (`red_extent[2]/2 + green_extent[2]/2`). This indicates a shift from memorization to grounded geometric reasoning.

**Zero-shot Generalization to Non-privileged Physical World Setups.** Although the model is only post-trained on the privileged S1 tier, the resulting policies demonstrate robust zero-shot transfer to the non-privileged S2 tier. Consequently, the model functions effectively in physical real-world setups, as shown in Figure 19.

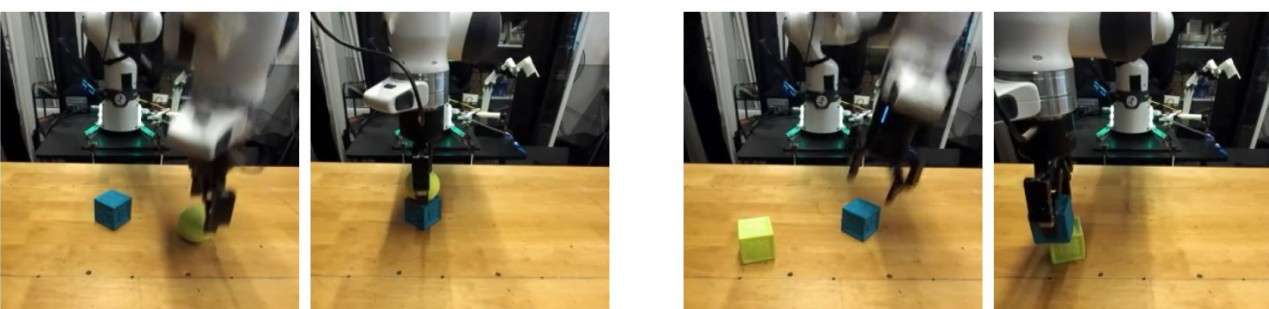

"Place the tennis ball on top of the blue cube" | "Place the blue cube on top of the yellow cube"

*Figure 20.* CaP-RL Finetuned Qwen-2.5-Coder-7B-Instruct successfully deployed on similar task variants. Demonstrates that base model is still capable of instruction following.

**Zero-Shot Generalization to Task Variants.** We observe instruction-following and zero-shot generalization to similar task variants after RL post-training, as seen in 20. The same policy successfully executes variants such as *"put the tennis ball on the green cube"* or tasks involving randomized object colors, as the underlying logic relies on abstract properties (bounding boxes) rather than overfitting to specific entity names or training instance scales.

# F. CaP-Agent0 Case Studies

## F.1. Real-World Case Studies

### F.1.1. COMMON SENSE PHYSICS-AWARE TASK DECOMPOSITION

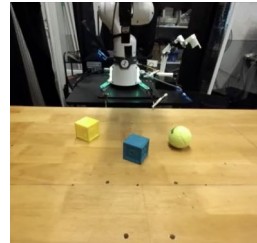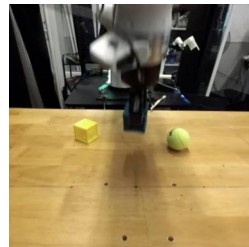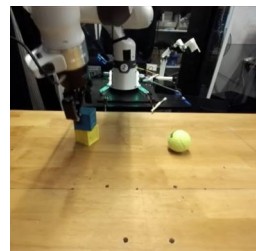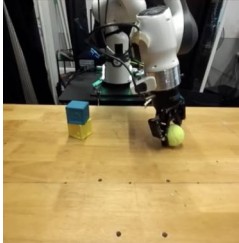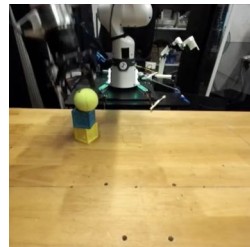

*Figure 21.* "Stack these as high as you can."

We validate CaP-Agent0 on a real-world Franka Panda platform using a set of geometrically heterogeneous objects: a yellow cube, a blue cube, and a tennis ball. The agent was given the open-ended instruction: *"stack these as high as you can."* with no additional prompt context outside of API usage documentation and the multimodal agent initial scene description.

Crucially, the task prompt provides neither a specific stacking sequence nor *any* semantic description of the scene or its objects. While a standard visuomotor policy might attempt to stack objects in detection order, potentially attempting to balance a cube on top of the spherical ball, CaP-Agent0 demonstrated common-sense embodied physical reasoning by deriving the only stable construction order.

This behavior is enabled by the CaP-Agent0 auxiliary VDM multimodal agent that extracts task-relevant image information before code generation begins. Prompted to "describe the initial state of the environment with the goal of the task in mind," the VLM emits the following context for the coding agent's initial turn:

> *"...The cubes appear to have flat surfaces suitable for stacking, while the tennis ball is spherical and would likely need to be placed on top or handled carefully."*

Leveraging this textual grounding, the coding agent explicitly decomposed the problem based on stability constraints. The full code generated by the CaP-Agent0 agent for completing this task is found in Appendix I.1.

### F.1.2. IMPLICIT MULTI-STEP REASONING FOR OBSTRUCTED GOALS

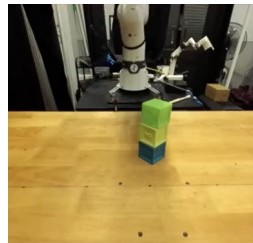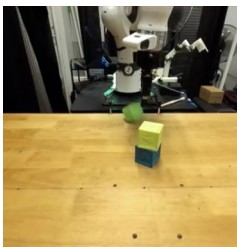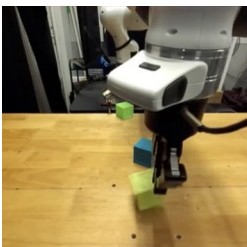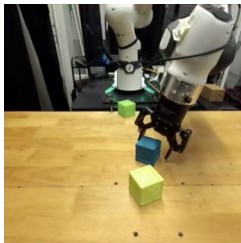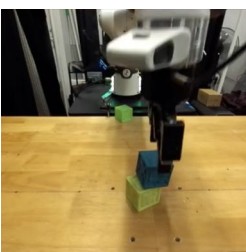

*Figure 22.* "Place the blue cube on top of the yellow cube."

We evaluate *CaP-Agent0* on a task where the *Franka Panda* must place a blue cube on top of a yellow cube. In the initial state, the cubes form a three-object tower: blue is at the base, supporting yellow, which in turn supports a green cube (Figure 22).

The goal prompt *"place the blue cube on top of the yellow cube"* is linguistically underspecified; it contains no mention of the green obstruction nor the fact that the blue target currently supports the yellow destination.

CaP-Agent0 resolves this dependency through its auxiliary multimodal perception layer. Before code generation, the VLM provides the coding agent with a high-level strategic prior:

*"...the blue cube is at the bottom of the stack, underneath the yellow cube, and the yellow cube is underneath a green cube. This implies a complex manipulation task where the robot will likely need to: 1. Remove the top (green) cube. 2. Remove the middle (yellow) cube to access the blue one, or move the stack. 3. Ultimately re-arrange them such that the blue cube ends up on top of the yellow one. The straightforward description is that the target blue cube is currently obstructed and supporting the target destination object (the yellow cube)."*

This behavior demonstrates that programmatic agents, when augmented with a dedicated perception-to-text grounding layer, can derive complex manipulation strategies traditionally requiring extensive task-specific imitation learning data or manual state-machine design. The full executable rollout is provided in Appendix I.2.

### F.2. Simulation Case Studies

#### F.2.1. VERIFICATION AND RETRY BEHAVIOR ENABLED BY MULTITURN DESIGN

This case study comes from the "pick up the red cube" task from Robosuite ported into CaP-Gym. Unlike one-shot code generation frameworks, the multi-turn architecture of CaP-Agent0 allows for emergent agentic resilience. In several simulation trials, we observed instances where a planned grasp was physically executed but failed to successfully lift the object due to physics-engine stochasticity or sub-optimal contact points. Rather than terminating the task upon failure, the agent utilized the subsequent turn's observation to verify that the goal condition (e.g., $z_{object} > z_{table}$) remained unfulfilled. As shown in the generated logic, the agent autonomously synthesized a recovery routine branch. This capacity to reason about execution failures and reset the environment state demonstrates that CaP-Agent0 can function as a closed-loop reactive agent rather than a static script generator.

```python
    ...

masks = segment_sam3_text_prompt(rgb, "red cube")

is_lifted = False
if not masks:
    print("Could not find red cube to verify state. Assuming failed and moving to home pose.")
    is_lifted = False
else:
    best_mask = max(masks, key=lambda x: x["score"])["mask"]

    y_idxs, x_idxs = np.where(best_mask)
    if len(y_idxs) > 0:
        stride = 10
        y_sample = y_idxs[::stride]
        x_sample = x_idxs[::stride]

        z_sample = depth[y_sample, x_sample]
        if z_sample.ndim > 1: z_sample = z_sample.flatten()

        # Filter invalid depth
        valid = z_sample > 0
        if np.any(valid):
            z_sample = z_sample[valid]
            x_sample = x_sample[valid]
            y_sample = y_sample[valid]

            fx = intrinsics[0, 0]
            fy = intrinsics[1, 1]
            cx = intrinsics[0, 2]
            cy = intrinsics[1, 2]

            x_c = (x_sample - cx) * z_sample / fx
            y_c = (y_sample - cy) * z_sample / fy

            points_c = np.stack([x_c, y_c, z_sample], axis=1)
            points_c_homo = np.hstack([points_c, np.ones((len(points_c), 1))])
            points_w = (cam_pose @ points_c_homo.T).T[:, :3]

            mean_z = np.mean(points_w[:, 2])
            print(f"Estimated mean Z of object: {mean_z:.4f}")

            # If Z > 0.15, we assume it's lifted (table usually ~0, cube size ~0.05)
```

```
44            if mean_z > 0.15:
45                is_lifted = True
46                print("Object is confirmed to be lifted.")
47            else:
48                print("Object appears to be on the table.")
49
50 if is_lifted:
51     print("Task completed successfully.")
52 else:
53     print("Retrying grasp sequence...")
54     open_gripper()
55
56     # Move to home pose to get a better view (avoid occlusion from gripper)
57     home_pos = np.array([0.5, 0.0, 0.5])
58     home_quat = np.array([0.0, 1.0, 0.0, 0.0]) # w,x,y,z (Approx gripper pointing down)
59
60     move_to_joints(solve_ik(home_pos, home_quat))
61
62     # Re-perceive
63     obs = get_observation()
64     rgb = obs["robot0_robotview"]["images"]["rgb"]
65     depth_im = obs["robot0_robotview"]["images"]["depth"]
66     intrinsics = obs["robot0_robotview"]["intrinsics"]
67     cam_pose = obs["robot0_robotview"]["pose_mat"]
68
69     masks = segment_sam3_text_prompt(rgb, "red cube")
70     if not masks:
71         print("No red cube found after reset.")
72     else:
73         best_mask = max(masks, key=lambda x: x["score"])["mask"]
74         if depth_im.ndim == 3: depth_map = depth_im[:, :, 0]
75         else: depth_map = depth_im
76
77         grasp_poses, grasp_scores = plan_grasp(depth_map, intrinsics, best_mask.astype(
   np.int32))
78
79         if len(grasp_poses) == 0:
80             print("No grasps found.")
81         else:
82             best_idx = np.argmax(grasp_scores)
83             best_pose_cam = grasp_poses[best_idx]
84
85             pose_world = cam_pose @ best_pose_cam
86             target_pos = pose_world[:3, 3]
87             target_rot = pose_world[:3, :3]
88             target_quat = rot_to_quat(target_rot)
89
90             # Grasp execution
91             # 1. Move to Pre-grasp
92             pre_pos = target_pos + np.array([0, 0, 0.1])
93             move_to_joints(solve_ik(pre_pos, target_quat))
94
95             # 2. Move to Grasp
96             move_to_joints(solve_ik(target_pos, target_quat))
97
98             # 3. Close
99             close_gripper()
100
101            # 4. Lift
102            lift_pos = target_pos + np.array([0, 0, 0.25])
103            move_to_joints(solve_ik(lift_pos, target_quat))
104            print("Retry finished.")
105
106     ...
```

### F.2.2. PERCEPTION MODULE CASCADE FAILURE FALLBACKS

This case study comes from the "wipe up the brown spill" task from Robosuite ported into CaP-Gym. The model frequently constructs robust perception pipelines that utilize hierarchical fallbacks. It understands that a generic text prompt might fail or be imprecise. In several trials, the model employs a VLM (Molmo) to visually ground the target ("brown spill") into a specific 2D point, and then uses that point to prompt a segmentation model (SAM3) for a precise mask. If this high-precision

method fails, it falls back to broader text prompts like "stain" or "brown object".

```
1  ...
2
3  # 2. Try detection using Molmo (VLM) for better semantic understanding
4  print("Attempting to detect spill using Molmo...")
5  molmo_res = point_prompt_molmo(rgb, "brown spill")
6  # Molmo returns dict[str, tuple], getting the first/only value
7  spill_pt = None
8  if molmo_res:
9      # Check if we got a valid coordinate
10     for key, pt in molmo_res.items():
11         if pt[0] is not None and pt[1] is not None:
12             spill_pt = pt
13             print(f"Molmo found spill at: {spill_pt}")
14             break
15 masks = []
16 if spill_pt:
17     # Use the point to segment with SAM3
18     print(f"Segmenting with SAM3 point prompt at {spill_pt}")
19     masks = segment_sam3_point_prompt(rgb, spill_pt)
20 else:
21     # Fallback to text prompt with different wording if Molmo fails
22     print("Molmo failed. Trying SAM3 text prompt 'stain'...")
23     masks = segment_sam3_text_prompt(rgb, "stain")
24     if not masks:
25         print("Fallback to 'brown liquid'...")
26         masks = segment_sam3_text_prompt(rgb, "brown liquid")
27 if not masks:
28     print("No masks found after all attempts. Cannot proceed.")
29     return
30 ...
```

### F.2.3. GEOMETRIC REASONING FOR GRASP SELECTION

This case study illustrates how CaP-Agent0 enhances sub-symbolic planners by injecting context-aware strategy. While modules like *Contact-GraspNet* infer many 6D grasp poses along with confidence scores, the learned scoring lacks awareness of the broader task setting. For this *tabletop* scenario, the agent correctly identified that a top-down approach strategy is inherently more reliable than side-approaches, regardless of their raw grasp scores.

To enforce this strategy, the agent autonomously synthesized a geometric constraint wrapper. By computing the alignment between grasp approach vectors and the world vertical, the generated code filtered out high-scoring but suboptimal side-grasps, explicitly selecting a vertical candidate to maximize stability. The agent also synthesized an autonomous fallback branch: if this strict strategy yielded no candidates, the logic reverted to the global maximum score to prevent execution stagnation. This demonstrates the agent's capacity to wrap generic tools in context-specific logic, bridging the gap between raw geometric perception and high-level task semantics.

Crucially, the automatically synthesized skill library (Appendix H.1) contains an analogous grasp filter, serving as a primary example of the agent's capacity to synthesize task-agnostic geometric logic into its persistent codebase.

```
1  ...
2
3  # 2. Grasp Planning
4  print("Planning grasp...")
5  grasp_poses, grasp_scores = plan_grasp(depth, intrinsics, seg_map)
6  if len(grasp_poses) == 0:
7      print("Error: No grasp poses found.")
8      return
9
10 # 3. Filter and Selection
11 # Transform all grasps to world frame to check orientation
12 best_score = -1.0
13 best_grasp_world = None
14 # We iterate through high-scoring grasps
15 indices = np.argsort(grasp_scores)[::-1] # Descending order
16 print(f"Found {len(indices)} candidates. Evaluating top 10...")
17 for idx in indices[:20]:
18     grasp_camera = grasp_poses[idx]
19     grasp_world = transform_camera_to_world(grasp_camera, camera_pose)
```

```
20       # Check approach vector (Z-axis of rotation matrix)
21       # We want the gripper to approach somewhat downwards (negative Z in world)
22       rot_mat = grasp_world[:3, :3]
23       approach_vec = rot_mat[:, 2] # The Z-axis
24       # Dot product with world down vector (0, 0, -1)
25       # If dot product is positive (angle < 90), it's pointing somewhat down.
26       # Actually, if approach is (0,0,-1), dot with (0,0,-1) is 1.
27       # We want approach_z to be negative.
28       if approach_vec[2] < -0.5: # At least 60 degrees downward tilt
29           print(f"Candidate {idx}: Score {grasp_scores[idx]:.3f}, Approach Z: {
     approach_vec[2]:.3f} (Good)")
30           best_grasp_world = grasp_world
31           break
32       else:
33           # print(f"Candidate {idx}: Score {grasp_scores[idx]:.3f}, Approach Z: {
     approach_vec[2]:.3f} (Skipping, too horizontal)")
34           pass
35   if best_grasp_world is None:
36       print("Warning: No steep top-down grasp found. Falling back to highest score.")
37       best_grasp_world = transform_camera_to_world(grasp_poses[indices[0]],
     camera_pose)
38   target_pos = best_grasp_world[:3, 3]
39   target_rot = best_grasp_world[:3, :3]
40   target_quat = rotation_matrix_to_quaternion(target_rot)
41   ...
```

### F.2.4. CODE GENERATION ENSEMBLING INCREASES ROBUSTNESS

In this case study, we analyze the differences in code generation for a VDM agent which uses a single query vs a model ensemble. All generated code is for the cube restack task. The following is a portion of the initial generation from single query.

```
1    ...
2    import numpy as np
3
4    # Step 1: Unstack the green cube from the red cube.
5    # Get grasp pose for the green cube (currently on top)
6    green_grasp_pos, green_grasp_quat = sample_grasp_pose('green cube')
7
8    # Approach and pick up the green cube
9    open_gripper()
10   goto_pose(green_grasp_pos, green_grasp_quat, z_approach=0.1)
11   close_gripper()
12   ...
```

Notice how there is no fallback for the `sample_grasp_pose()` generation. When the initial code was executed, the outputted grasp was invalid. The model reasoned that the grasp was invalid because it was too close to the origin, so in the subsequent generation, the model retroactively implemented a fallback which only prevents the failure case it just encountered.

```
1    ...
2    # Based on the logs, the previous execution seems to have proceeded through the
     logic,
3    # but looking at the sampled grasp pose for the red cube: [ 0.04235514  0.08625826
     -0.04337643]
4    # compared to the get_object_pose for red cube: [ 0.68666898  0.12055038
     -0.09239209]
5    # The sampled grasp pose for the red cube seems completely wrong (close to robot
     base origin 0,0,0).
6    # This likely caused the robot to fail picking up the red cube or crash/move
     incorrectly in Step 2.
7    ...
8    # Get poses
9    red_pos, _, _ = get_object_pose('red cube')
10   green_pos, _, green_bbox = get_object_pose('green cube', return_bbox_extent=True)
11
12   # Sample grasp for red cube
13   red_grasp_pos, red_grasp_quat = sample_grasp_pose('red cube')
14
```

```
15      # Sanity check: if grasp pose is near 0,0,0, fall back to object pose + offset
16      if np.linalg.norm(red_grasp_pos) < 0.2:
17          # Fallback: use object position directly for grasp center, oriented top-down
18          print("Warning: Sampled grasp pose seems invalid. Using object position
        fallback.")
19          red_grasp_pos = red_pos.copy()
20          red_grasp_quat = np.array([0, 0, 1, 0]) # Top down for fallback
21      ...
```

On the other hand, generations from a model ensemble tend to anticipate failure cases and preemptively implement fallbacks as shown through the following initial generation:

```
1      ...
2      def pick_object(mask, depth, intrinsics, extrinsics, z_lift=0.2):
3          """Plans and executes a grasp for the object defined by the mask."""
4          # Plan grasp
5          grasps, scores = plan_grasp(depth, intrinsics, mask.astype(np.int32))
6          best_grasp_world, _ = select_top_down_grasp(grasps, scores, extrinsics,
        vertical_threshold=0.7)
7
8          if best_grasp_world is None:
9              # Fallback to centroid grasp if grasp planner fails
10             print("Grasp planner failed, falling back to centroid grasp.")
11             stats = get_object_stats(mask, depth, intrinsics, extrinsics)
12             if stats is None:
13                 raise RuntimeError("Cannot compute object stats for fallback grasp.")
14
15             # Create a default top-down orientation
16             R_down = np.array([[1.0, 0.0, 0.0], [0.0, -1.0, 0.0], [0.0, 0.0, -1.0]])
17             quat = rotation_matrix_to_quaternion(R_down)
18             pos = stats["center"].copy()
19             pos[2] = stats["max_z"] - 0.02  # Grasp slightly below top
20         else:
21             pos, quat = decompose_transform(best_grasp_world)
22
23         # Execute sequence
24         pre_grasp = pos.copy()
25         pre_grasp[2] += 0.12  # 12cm above
26
27         open_gripper()
28         move_tcp(pre_grasp, quat)
29         move_tcp(pos, quat)
30     ...
```

# G. Low-Level Perception and Control Primitives

This section provides the complete specifications for the low-level perception and control primitives used in CaP-Bench, including function signatures, interface definitions, and documentation strings. These primitives constitute the exact specifications provided to agents in Tier S3. Tier S4 utilizes an identical primitive set, but with in-context usage examples (the Example: sections within docstrings) stripped. Tiers S1 and S2 operate on high-level abstractions that are constructed by composing these fundamental primitives.

```
1  from typing import Any
2  import numpy as np
3  import open3d as o3d
4  import viser.transforms as vtf
5  from PIL import Image
6  from scipy.spatial.transform import Rotation as SciRotation
7  from capx.envs.base_env import BaseEnv
8  from capx.integrations import pyroki_snippets as pks  # type: ignore
9  from capx.integrations.base_api import ApiBase
10 from capx.integrations.grasp_graspnet import init_contact_graspnet
11 from capx.integrations.molmo import init_molmo
12 from capx.integrations.pyroki import init_pyroki
13 from capx.integrations.sam3 import init_sam3, init_sam3_point_prompt
14
15 class S3(ApiBase):
16     """
17     Robot perception and control primitives for the S3 CaP-Bench Tier
```

```python
18      """
19      def __init__(
20          self,
21          env: BaseEnv,
22          tcp_offset: list[float] | None = [0.0, 0.0, -0.107],
23          bimanual: bool = False,
24      ) -> None:
25          super().__init__(env)
26          self._TCP_OFFSET = np.array(tcp_offset, dtype=np.float64)
27          self.grasp_net_plan_fn = init_contact_graspnet()
28          self.sam3_seg_fn = init_sam3()
29          self.sam3_point_prompt_fn = init_sam3_point_prompt()
30          self.molmo_point_fn = init_molmo()
31
32          self.ik_solve_fn = init_pyroki()
33          self.trajopt_plan_fn = init_pyroki_trajopt()
34          self.cfg = None
35          self.bimanual = bimanual
36
37      def functions(self) -> dict[str, Any]:
38          fns = {
39              "get_observation": self.get_observation,
40              "segment_sam3_text_prompt": self.segment_sam3_text_prompt,
41              "segment_sam3_point_prompt": self.segment_sam3_point_prompt,
42              "point_prompt_molmo": self.point_prompt_molmo,
43              "plan_grasp": self.plan_grasp,
44              "get_oriented_bounding_box_from_3d_points": self.
      get_oriented_bounding_box_from_3d_points,
45          }
46          if self.bimanual:
47              fns["solve_ik_arm0"] = self.solve_ik_arm0
48              fns["solve_ik_arm1"] = self.solve_ik_arm1
49              fns["move_to_joints_both"] = self.move_to_joints_both
50              fns["move_to_joints_arm0"] = self.move_to_joints_arm0
51              fns["move_to_joints_arm1"] = self.move_to_joints_arm1
52              fns["open_gripper_arm0"] = self.open_gripper_arm0
53              fns["close_gripper_arm0"] = self.close_gripper_arm0
54              fns["open_gripper_arm1"] = self.open_gripper_arm1
55              fns["close_gripper_arm1"] = self.close_gripper_arm1
56          else:
57              fns["solve_ik"] = self.solve_ik
58              fns["move_to_joints"] = self.move_to_joints
59              fns["open_gripper"] = self.open_gripper
60              fns["close_gripper"] = self.close_gripper
61
62          return fns
63
64      def get_observation(self) -> dict[str, Any]:
65          """Get the observation of the environment.
66          Returns:
67              observation:
68                  A dictionary containing the observation of the environment.
69                  The dictionary contains the following keys:
70                  - ["robot0_robotview"]["images"]["rgb"]: Current color camera image as
      a numpy array of shape (H, W, 3), dtype uint8.
71                  - ["robot0_robotview"]["images"]["depth"]: Current depth camera image
      as a numpy array of shape (H, W, 1), dtype float32.
72                  - ["robot0_robotview"]["intrinsics"]: Camera intrinsic matrix as a
      numpy array of shape (3, 3), dtype float64.
73                  - ["robot0_robotview"]["pose_mat"]: Camera extrinsic matrix as a numpy
      array of shape (4, 4), dtype float64.
74
75          """
76          return self._env.get_observation()
77
78      # ----------------------------------------------------------------------- #
79      # Vision models: Sam3 segmentation
80      # ----------------------------------------------------------------------- #
81
82      def segment_sam3_point_prompt(
83          self,
84          rgb: np.ndarray,
85          point_coords: tuple[float, float],
```

```
 86      ) -> list[dict[str, Any]]:
 87          """Run SAM3 segmentation on an RGB image, optionally conditioned on an image
     coordinate point prompt.
 88
 89          Args:
 90              rgb:
 91                  RGB image array of shape (H, W, 3), dtype uint8.
 92              point_coords:
 93                  (x, y) pixel coordinates of the point prompt.
 94
 95          Returns:
 96              masks:
 97                  A list of dictionaries. Each dict may contain:
 98
 99                      - "mask":  np.ndarray of shape (H, W), dtype bool or uint8,
100                                 where True/1 means the pixel belongs to the instance.
101                      - "score": float confidence score.
102
103          Example:
104              >>> rgb = obs["robot0_robotview"]["images"]["rgb"]
105              >>> masks = segment_sam3_point_prompt(rgb, (100, 100))
106          """
107          return self.sam3_point_prompt_fn(Image.fromarray(rgb), point_coords)
108
109      def segment_sam3_text_prompt(
110          self,
111          rgb: np.ndarray,
112          text_prompt: str,
113      ) -> list[dict[str, Any]]:
114          """Run SAM3 segmentation on an RGB image conditioned on a text prompt.
115
116          Args:
117              rgb:
118                  RGB image array of shape (H, W, 3), dtype uint8.
119              text_prompt:
120                  Text prompt for SAM3 segmentation.
121
122          Returns:
123              masks:
124                  A list of dictionaries. Each dict may contain:
125
126                      - "mask":  np.ndarray of shape (H, W), dtype bool or uint8,
127                                 where True/1 means the pixel belongs to the instance.
128                      - "box": list [x1, y1, x2, y2] in pixel coordinates.
129                      - "score": float confidence score.
130
131          Example:
132              >>> rgb = obs["robot0_robotview"]["images"]["rgb"]
133              >>> masks = segment_sam3(rgb, text_prompt="red mug")
134          """
135          return self.sam3_seg_fn(rgb, text_prompt=text_prompt)
136
137      # ---------------------------------------------------------------------- #
138      # Molmo point prompt
139      # ---------------------------------------------------------------------- #
140      def point_prompt_molmo(
141          self,
142          image: np.ndarray,
143          text_prompt: str,
144      ) -> dict[str, tuple[int | None, int | None]]:
145          """Use Molmo to point to a coordinate in the image based on a text prompt.
146
147          Args:
148              image: np.ndarray: The RGB image to process. Shape: (H, W, 3), dtype uint8.
149              text_prompt: str: The text prompt to point to.
150
151          Returns:
152              dict[str, tuple[int | None, int | None]]: Pixel coordinates for each
153              object query; (None, None) if parsing failed.
154          """
155          return self.molmo_point_fn(Image.fromarray(image), objects=[text_prompt])
156
157      def get_oriented_bounding_box_from_3d_points(self, points: np.ndarray) -> dict[str,
```

```
        Any]:
158         """Get the oriented bounding box from 3D points.
159
160         Args:
161             points: np.ndarray: The 3D points to get the oriented bounding box from.
162                 Shape: (N, 3), dtype float64.
163
164         Returns:
165             dict[str, Any]: The oriented bounding box. The dictionary contains the
        following keys:
166                 - "center": np.ndarray: The center of the oriented bounding box in
        point cloud frame.
167                 - "extent": np.ndarray: The extent of the oriented bounding box.
168                 - "R": np.ndarray: The rotation matrix of the oriented bounding box in
        point cloud frame.
169         """
170         o3d_points = o3d.geometry.PointCloud()
171         o3d_points.points = o3d.utility.Vector3dVector(points)
172         o3d_points, ind = o3d_points.remove_statistical_outlier(nb_neighbors=20,
        std_ratio=2.0)
173         obb = o3d_points.get_oriented_bounding_box()
174         return {
175             "center": obb.center,
176             "extent": obb.extent,
177             "R": obb.R,
178         }
179
180     # ---------------------------------------------------------------------- #
181     # Grasp planner (Contact-GraspNet)
182     # ---------------------------------------------------------------------- #
183     def plan_grasp(
184         self,
185         depth: np.ndarray,
186         intrinsics: np.ndarray,
187         segmentation: np.ndarray,
188     ) -> tuple[np.ndarray, np.ndarray]:
189         """Plan grasp candidates using Contact-GraspNet for a single instance.
190
191         This is a thin wrapper around the Contact-GraspNet planner. It does not
192         apply any camera/world transforms or TCP offsets: the caller is
193         responsible for transforming the resulting grasp poses into the desired
194         frame and applying TCP offsets if necessary.
195
196         Args:
197             depth:
198                 Depth image in meters.
199                 Shape: (H, W) or (H, W, 1), dtype float32/float64.
200             intrinsics:
201                 Camera intrinsic matrix.
202                 Shape: (3, 3), dtype float64.
203             segmentation:
204                 Instance segmentation map where each integer > 0 corresponds to a
205                 unique object instance ID.
206                 Shape: (H, W) or (H, W, 1), dtype int32/int64.
207
208         Returns:
209             grasp_poses:
210                 np.ndarray of shape (K, 4, 4), dtype float64.
211                 Homogeneous transforms for each candidate grasp IN THE CAMERA FRAME.
212             grasp_scores:
213                 np.ndarray of shape (K,), dtype float64.
214                 Confidence score for each candidate grasp.
215
216         Example:
217             >>> cam = obs["robot0_robotview"]
218             >>> rgb = cam["images"]["rgb"]
219             >>> depth = cam["images"]["depth"][:, :, 0]
220             >>> sam3_results = sam3_seg_fn(rgb, text_prompt="red mug")
221             >>> best = max(sam3_results, key=lambda d: d["score"])
222             >>> mask = best["mask"]
223             >>> K = cam["intrinsics"]
224             >>> grasp_sample_tf, grasp_scores = plan_grasp(
225             ...     depth=depth,
```

```
226              ...        intrinsics=K,
227              ...        segmentation=mask,
228              ... )
229              >>> best_idx = grasp_scores.argmax()
230              >>> best_T = grasp_poses[best_idx]  # (4, 4)
231              >>> camera_extrinsics = cam["pose_mat"]
232              >>> grasp_sample_world_frame = camera_extrinsics @ best_T
233          """
234          if depth.ndim == 3 and depth.shape[-1] == 1:
235              depth = depth[:, :, 0]
236          if segmentation.ndim == 3 and segmentation.shape[-1] == 1:
237              segmentation = segmentation[:, :, 0]
238
239          grasp_sample, grasp_scores, _ = self.grasp_net_plan_fn(
240              depth,
241              intrinsics,
242              segmentation,
243              1,
244              z_range=[0.2, 3.5] if self.is_handover else [0.2, 2.0],
245              forward_passes=1 if self.is_handover else 3,
246          )
247
248          grasp_sample_tf = (
249              vtf.SE3.from_matrix(grasp_sample) @ vtf.SE3.from_translation(np.array([0,
     0, 0.12]))
250          ).as_matrix()
251
252          return grasp_sample_tf, grasp_scores
253
254      # ---------------------------------------------------------------------- #
255      # IK / motion primitives
256      # ---------------------------------------------------------------------- #
257      def solve_ik(
258          self,
259          position: np.ndarray,
260          quaternion_wxyz: np.ndarray,
261      ) -> np.ndarray:
262          """Solve inverse kinematics for the panda_hand link.
263
264          Args:
265              position:
266                  Target position in world frame.
267                  Shape: (3,), dtype float64.
268              quaternion_wxyz:
269                  Target orientation as a unit quaternion in world frame.
270                  Shape: (4,), [w, x, y, z], dtype float64.
271
272          Returns:
273              joints:
274                  np.ndarray of shape (7,), dtype float64.
275                  Joint angles for the 7 DoF Franka arm.
276
277          Example:
278              >>> target_pos = np.array([0.5, 0.0, 0.3])
279              >>> target_quat = np.array([1.0, 0.0, 0.0, 0.0])  # identity, wxyz
280              >>> joints = solve_ik(target_pos, target_quat)
281              >>> move_to_joints(joints)
282          """
283          pos = np.asarray(position, dtype=np.float64).reshape(3)
284          quat_wxyz = np.asarray(quaternion_wxyz, dtype=np.float64).reshape(4)
285          quat_xyzw = np.array(
286              [quat_wxyz[1], quat_wxyz[2], quat_wxyz[3], quat_wxyz[0]], dtype=np.float64
287          )
288          rot = SciRotation.from_quat(quat_xyzw)
289          offset_pos = pos + rot.apply(self._TCP_OFFSET)
290
291          prev_cfg = self.cfg
292
293          for i in range(15):  # run w/ multiple iterations when using vel_cost ik solver
294              self.cfg = self.ik_solve_fn(
295                  target_pose_wxyz_xyz=np.concatenate([quat_wxyz, offset_pos]),
296                  prev_cfg=prev_cfg,
297              )
```

```
298                if prev_cfg is not None:
299                    if np.allclose(self.cfg, prev_cfg, atol=1e-3):
300                        break
301                    else:
302                        prev_cfg = self.cfg
303            joints = np.asarray(self.cfg[:-1], dtype=np.float64).reshape(7)
304            return joints
305
306        # Single arm control APIs
307
308        def move_to_joints(self, joints: np.ndarray) -> None:
309            """Move the robot to a given joint configuration in a blocking manner.
310
311            Args:
312                joints:
313                    Target joint angles for the 7-DoF Franka arm.
314                    Shape: (7,), dtype float64.
315
316            Returns:
317                None
318
319            Example:
320                >>> joints = np.array([0.0, -0.5, 0.0, -2.0, 0.0, 1.5, 0.8])
321                >>> move_to_joints(joints)
322            """
323            joints = np.asarray(joints, dtype=np.float64).reshape(7)
324            self._env.move_to_joints_blocking(joints)
325
326        def open_gripper(self) -> None:
327            """Open gripper fully.
328
329            Args:
330                None
331            """
332            self._env._set_gripper(1.0)
333            for _ in range(30):
334                self._env._step_once()
335
336        def close_gripper(self) -> None:
337            """Close gripper fully.
338
339            Args:
340                None
341            """
342            self._env._set_gripper(0.0)
343            for _ in range(30):
344                self._env._step_once()
345
346
347        # Dual arm control APIs
348        def move_to_joints_both(self, joints0: np.ndarray, joints1: np.ndarray) -> None:
349            """Move the arms 0 and 1 to a given joint configuration in a blocking manner
        simultaneously.
350
351            Args:
352                joints0:
353                    Target joint angles for the 7-DoF Franka arm 0.
354                    Shape: (7,), dtype float64.
355                joints1:
356                    Target joint angles for the 7-DoF Franka arm 1.
357                    Shape: (7,), dtype float64.
358            """
359            self._env.move_to_joints_blocking_both(joints0, joints1)
360
361        def move_to_joints_arm0(self, joints: np.ndarray) -> None:
362            """Move the robot arm 0 to a given joint configuration in a blocking manner.
363
364            Args:
365                joints:
366                    Target joint angles for the 7-DoF Franka arm 0.
367                    Shape: (7,), dtype float64.
368            """
369            joints = np.asarray(joints, dtype=np.float64).reshape(7)
```

```
370         self._env.move_to_joints_blocking(joints)

372     def move_to_joints_arm1(self, joints: np.ndarray) -> None:
373         """Move the robot arm 1 to a given joint configuration in a blocking manner.

375         Args:
376             joints:
377                 Target joint angles for the 7-DoF Franka arm 1.
378                 Shape: (7,), dtype float64.
379         """
380         joints = np.asarray(joints, dtype=np.float64).reshape(7)
381         self._env.move_to_joints_blocking_arm1(joints)

383     def open_gripper_arm0(self) -> None:
384         """Open gripper fully for Arm 0 (robot0).
385         Args:
386             None
387         Returns:
388             None
389         """
390         self._env._set_gripper(1.0)
391         for _ in range(30):
392             self._env._step_once()

394     def close_gripper_arm0(self) -> None:
395         """Close gripper fully for Arm 0 (robot0).
396         Args:
397             None
398         Returns:
399             None
400         """
401         self._env._set_gripper(0.0)
402         for _ in range(30):
403             self._env._step_once()
404     def open_gripper_arm1(self) -> None:
405         """Open gripper fully for Arm 1 (robot1).
406         Args:
407             None
408         Returns:
409             None
410         """
411         self._env._set_gripper_arm1(1.0)
412         for _ in range(30):
413             self._env._step_once()

415     def close_gripper_arm1(self) -> None:
416         """Close gripper fully for Arm 1 (robot1).
417         Args:
418             None
419         Returns:
420             None
421         """
422         self._env._set_gripper_arm1(0.0)
423         for _ in range(30):
424             self._env._step_once()

426     def solve_ik_arm0(self, position: np.ndarray, quaternion_wxyz: np.ndarray) -> np.
    ndarray:
427         """Solve inverse kinematics for the panda_hand link for Arm 0 (robot0)."""
428         pos = np.asarray(position, dtype=np.float64).reshape(3)
429         quat_wxyz = np.asarray(quaternion_wxyz, dtype=np.float64).reshape(4)
430         quat_xyzw = np.array(
431             [quat_wxyz[1], quat_wxyz[2], quat_wxyz[3], quat_wxyz[0]], dtype=np.float64
432         )
433         rot = SciRotation.from_quat(quat_xyzw)
434         offset_pos = pos + rot.apply(self._TCP_OFFSET)

436         prev_cfg = self.cfg
437         for i in range(15):
438             self.cfg = self.ik_solve_fn(
439                 target_pose_wxyz_xyz=np.concatenate([quat_wxyz, offset_pos]),
440                 prev_cfg=prev_cfg,
441             )
```

```
442            if prev_cfg is not None:
443                if np.allclose(self.cfg, prev_cfg, atol=1e-3):
444                    break
445                else:
446                    prev_cfg = self.cfg
447
448        joints = np.asarray(self.cfg[:-1], dtype=np.float64).reshape(7)
449        return joints
450
451    def solve_ik_arm1(self, position: np.ndarray, quaternion_wxyz: np.ndarray) -> np.
     ndarray:
452        """Solve inverse kinematics for the panda_hand link for Arm 1 (robot1)."""
453        if not hasattr(self._env, "move_to_joints_blocking_arm1"):
454            raise RuntimeError("Environment does not support Arm 1 control")
455
456        if not hasattr(self._env, "base_link_wxyz_xyz_0") or not hasattr(
457            self._env, "base_link_wxyz_xyz_1"
458        ):
459            raise RuntimeError("Environment does not provide base transforms.")
460
461        pose_arm0_base = vtf.SE3.from_rotation_and_translation(
462            rotation=vtf.SO3(wxyz=quaternion_wxyz),
463            translation=position,
464        )
465        base0_transform = vtf.SE3(wxyz_xyz=self._env.base_link_wxyz_xyz_0)
466        pose_world = base0_transform @ pose_arm0_base
467
468        base1_transform = vtf.SE3(wxyz_xyz=self._env.base_link_wxyz_xyz_1)
469        base1_transform_inv = base1_transform.inverse()
470        pose_arm1_base = base1_transform_inv @ pose_world
471
472        pos = np.asarray(pose_arm1_base.translation(), dtype=np.float64).reshape(3)
473        quat_wxyz = np.asarray(pose_arm1_base.rotation().wxyz, dtype=np.float64).
     reshape(4)
474        quat_xyzw = np.array(
475            [quat_wxyz[1], quat_wxyz[2], quat_wxyz[3], quat_wxyz[0]], dtype=np.float64
476        )
477        rot = SciRotation.from_quat(quat_xyzw)
478        offset_pos = pos + rot.apply(self._TCP_OFFSET)
479
480        prev_cfg = self.cfg
481        for i in range(15):
482            self.cfg = self.ik_solve_fn(
483                target_pose_wxyz_xyz=np.concatenate([quat_wxyz, offset_pos]),
484                prev_cfg=prev_cfg,
485            )
486            if prev_cfg is not None:
487                if np.allclose(self.cfg, prev_cfg, atol=1e-3):
488                    break
489                else:
490                    prev_cfg = self.cfg
491        joints = np.asarray(self.cfg[:-1], dtype=np.float64).reshape(7)
492        return joints
```

# H. CaP-Agent0 Details

### H.1. Synthesized Task-Agnostic Skill Library

In this section, we list all nine agent-synthesized function definitions, interfaces, documentation strings, and low-level implementations added to the CaP-Agent0 skill library.

```
1    def rotation_matrix_to_quaternion(R: np.ndarray) -> np.ndarray:
2        """
3        Convert a 3x3 rotation matrix to a unit quaternion [w, x, y, z].
4
5        Implements the robust Sheppard's method (checking trace and diagonal elements)
6        to avoid numerical instability when the trace is close to zero.
7        Args:
8            R: (3, 3) rotation matrix.
9        Returns:
10           np.array: [w, x, y, z] unit quaternion.
```

```
11          """
12          tr = np.trace(R)
13          if tr > 0:
14              S = np.sqrt(tr + 1.0) * 2
15              w = 0.25 * S
16              x = (R[2, 1] - R[1, 2]) / S
17              y = (R[0, 2] - R[2, 0]) / S
18              z = (R[1, 0] - R[0, 1]) / S
19          elif (R[0, 0] > R[1, 1]) and (R[0, 0] > R[2, 2]):
20              S = np.sqrt(1.0 + R[0, 0] - R[1, 1] - R[2, 2]) * 2
21              w = (R[2, 1] - R[1, 2]) / S
22              x = 0.25 * S
23              y = (R[0, 1] + R[1, 0]) / S
24              z = (R[0, 2] + R[2, 0]) / S
25          elif R[1, 1] > R[2, 2]:
26              S = np.sqrt(1.0 + R[1, 1] - R[0, 0] - R[2, 2]) * 2
27              w = (R[0, 2] - R[2, 0]) / S
28              x = (R[0, 1] + R[1, 0]) / S
29              y = 0.25 * S
30              z = (R[1, 2] + R[2, 1]) / S
31          else:
32              S = np.sqrt(1.0 + R[2, 2] - R[0, 0] - R[1, 1]) * 2
33              w = (R[1, 0] - R[0, 1]) / S
34              x = (R[0, 2] + R[2, 0]) / S
35              y = (R[1, 2] + R[2, 1]) / S
36              z = 0.25 * S
37          return np.array([w, x, y, z])
38
39      def decompose_transform(T: np.ndarray) -> tuple[np.ndarray, np.ndarray]:
40          """
41          Decompose a 4x4 homogeneous transformation matrix into position and quaternion.
42          Args:
43              T: (4, 4) homogeneous transformation matrix.
44          Returns:
45              tuple:
46                  - position: (3,) np.array
47                  - quaternion: (4,) np.array [w, x, y, z]
48          """
49          position = T[:3, 3]
50          R = T[:3, :3]
51          quat = rotation_matrix_to_quaternion(R)
52          return position, quat
53
54      def depth_to_point_cloud(depth_img: np.ndarray, intrinsics: np.ndarray) -> np.
    ndarray:
55          """
56          Convert a depth image to a 3D point cloud in the Camera Frame.
57          Args:
58              depth_img: (H, W) depth map in meters.
59              intrinsics: (3, 3) camera intrinsic matrix.
60          Returns:
61              np.array: (H, W, 3) image of 3D coordinates.
62          """
63          if depth_img.ndim == 3:
64              depth_img = depth_img[:, :, 0]
65
66          h, w = depth_img.shape
67          fx = intrinsics[0, 0]
68          fy = intrinsics[1, 1]
69          cx = intrinsics[0, 2]
70          cy = intrinsics[1, 2]
71
72          # Vectorized grid generation
73          y_grid, x_grid = np.mgrid[0:h, 0:w]
74
75          z = depth_img
76          x = (x_grid - cx) * z / fx
77          y = (y_grid - cy) * z / fy
78
79          return np.dstack((x, y, z))
80
81      def mask_to_world_points(
82          mask: np.ndarray, depth: np.ndarray, intrinsics: np.ndarray, extrinsics: np.
```

```python
     ndarray
83   ) -> np.ndarray:
84       """
85       Convert specific pixels defined by a binary mask into 3D points in the World
     Frame.
86       Args:
87           mask: (H, W) binary mask (0 or 1).
88           depth: (H, W) depth map.
89           intrinsics: (3, 3) camera intrinsics.
90           extrinsics: (4, 4) camera-to-world pose matrix.
91       Returns:
92           np.array: (N, 3) array of valid 3D points in world coordinates.
93       """
94       # Get pixel coordinates
95       ys, xs = np.where(mask > 0)
96       if len(ys) == 0:
97           return np.empty((0, 3))
98
99       z_vals = depth[ys, xs]
100
101      # Filter invalid depth
102      valid = z_vals > 0
103      ys = ys[valid]
104      xs = xs[valid]
105      z = z_vals[valid]
106
107      fx = intrinsics[0, 0]
108      fy = intrinsics[1, 1]
109      cx = intrinsics[0, 2]
110      cy = intrinsics[1, 2]
111
112      # Deproject to Camera Frame
113      x_cam = (xs - cx) * z / fx
114      y_cam = (ys - cy) * z / fy
115
116      # Stack to (N, 3)
117      points_cam = np.stack([x_cam, y_cam, z], axis=-1)
118
119      # Transform to World Frame
120      # Create homogeneous coordinates (N, 4)
121      points_cam_hom = np.hstack([points_cam, np.ones((len(points_cam), 1))])
122      points_world_hom = (extrinsics @ points_cam_hom.T).T
123
124      return points_world_hom[:, :3]
125
126  def pixel_to_world_point(
127      u: int, v: int, z: float, intrinsics: np.ndarray, extrinsics: np.ndarray
128  ) -> np.ndarray:
129      """
130      Deproject a single pixel to a 3D world point.
131      Args:
132          u, v: Pixel coordinates (col, row).
133          z: Depth at that pixel.
134          intrinsics: (3, 3) matrix.
135          extrinsics: (4, 4) matrix.
136      Returns:
137          np.array: [x, y, z] in world frame.
138      """
139      fx = intrinsics[0, 0]
140      fy = intrinsics[1, 1]
141      cx = intrinsics[0, 2]
142      cy = intrinsics[1, 2]
143
144      x_cam = (u - cx) * z / fx
145      y_cam = (v - cy) * z / fy
146
147      p_cam = np.array([x_cam, y_cam, z, 1.0])
148      p_world = extrinsics @ p_cam
149      return p_world[:3]
150
151  def transform_points(points: np.ndarray, transform_matrix: np.ndarray) -> np.
     ndarray:
152      """
```

```
153        Apply a 4x4 homogeneous transform to a set of 3D points.
154        Args:
155            points: (N, 3) or (H, W, 3) array of points.
156            transform_matrix: (4, 4) homogeneous transformation matrix.
157        Returns:
158            np.array: Transformed points with same shape as input.
159        """
160        original_shape = points.shape
161        # Flatten to (N, 3)
162        points_reshaped = points.reshape(-1, 3)
163
164        # Convert to homogeneous (N, 4)
165        ones = np.ones((points_reshaped.shape[0], 1))
166        points_hom = np.hstack((points_reshaped, ones))
167
168        # Apply transform: (4,4) @ (4,N) -> (4,N) -> Transpose back to (N,4)
169        points_transformed = (transform_matrix @ points_hom.T).T
170
171        # Return to (N, 3) and original shape
172        return points_transformed[:, :3].reshape(original_shape)
173
174    def interpolate_segment(
175        p1: np.ndarray, p2: np.ndarray, step: float = 0.03
176    ) -> list[np.ndarray]:
177        """
178        Generate waypoints along a line segment between two 3D points.
179        Args:
180            p1: Start point (3,).
181            p2: End point (3,).
182            step: Distance between waypoints in meters.
183        Returns:
184            list[np.ndarray]: List of points including p1 and p2.
185        """
186        dist = np.linalg.norm(p2 - p1)
187        if dist < 1e-6:
188            return [p1]
189        num_points = int(np.ceil(dist / step))
190        # Using linspace to ensure we hit the start and end exactly
191        return [p1 + (p2 - p1) * t for t in np.linspace(0, 1, num_points + 1)]
192
193    def normalize_vector(v: np.ndarray) -> np.ndarray:
194        """
195        Normalize a vector to unit length.
196        Args:
197            v: (3,) vector.
198        Returns:
199            np.array: (3,) unit vector.
200        """
201        norm = np.linalg.norm(v)
202        if norm < 1e-6:
203            return v
204        return v / norm
205
206    def select_top_down_grasp(
207        grasps: np.ndarray,
208        scores: np.ndarray,
209        cam_to_world: np.ndarray,
210        vertical_threshold: float = 0.8,
211    ) -> tuple:
212        """
213        Selects the best grasp that aligns the gripper vertically (Top-Down).
214        Args:
215            grasps: (N, 4, 4) Grasp poses in camera frame.
216            scores: (N,) Grasp scores.
217            cam_to_world: (4, 4) Extrinsics matrix.
218            vertical_threshold: Dot product threshold (1.0 is perfectly vertical).
219        Returns:
220            tuple: (best_grasp_world_matrix, best_score) or (None, -inf)
221        """
222        best_grasp = None
223        best_score = -np.float64("inf")
224
225        # World Z axis (vertical)
```

```
226        world_z = np.array([0, 0, 1])
227
228        for i, g_camera in enumerate(grasps):
229            # Transform grasp to world frame
230            g_world = cam_to_world @ g_camera
231
232            # Extract rotation
233            R = g_world[:3, :3]
234
235            # Assuming Gripper Z or Y is the approach vector depending on gripper
    definition.
236            # For Franka/Robotiq, the approach vector is usually the Z-axis of the end
    effector.
237            gripper_approach = R[:, 2]
238
239            # Check alignment with negative World Z (pointing down)
240            # Dot product should be close to -1 for top-down
241            alignment = -np.dot(gripper_approach, world_z)
242
243            if alignment > vertical_threshold:
244                if scores[i] > best_score:
245                    best_score = scores[i]
246                    best_grasp = g_world
247
248        return best_grasp, best_score
```

## H.2. Model Ensemble Temperature Details

In both single and multi-model settings, 9 candidates responses are generated. For single model, we query Gemini-3-Pro 9 times with temperatures 0.1, 0.2, 0.3,..., 0.9. For multi-model, we query Gemini-3-Pro, Claude-Opus-4.5, and GPT-5.2 3 times each with temperatures 0.1, 0.5, and 0.9.

## H.3. Model Ensemble Prompt

This section contains the prompts the coding agent uses to generate the final solution from the candidates for both the initial generation and subsequent multi-turn attempts. Text inside curly braces "{}" represent fstring placeholders.

User prompt for both initial and subsequent generations:

```
1    Synthesize the best solution.
2
3    <original_task_description>
4    {original prompt for candidates}
5    </original_task_description>
6
7    <candidate_solutions>
8    {candidate generations}
9    </candidate_solutions>
```

System prompt for initial generation:

```
1    You are synthesizing {# of candidate generations} candidate Python solutions into
     one optimal program.
2
3    SYNTHESIS RULES:
4    1. Analyze critically and assume no candidate is fully correct
5    2. Prefer explicit checks over assumptions
6    3. Combine the best ideas from multiple candidates when appropriate
7    4. If candidates disagree fundamentally, choose the more robust approach
8
9    OUTPUT FORMAT (strict):
10   You may include reasoning before the fenced code block.
11   Output ONLY ONE fenced code block (```python...```) containing the complete final
     solution.
12   Do NOT include any other code blocks or code snippets outside this single block.
```

System prompt for subsequent generations:

```
1    You are synthesizing {# of generations} candidate responses for a multi-turn robot
     control task.
2
3    DECISION ANALYSIS:
4    - {regenerate_count} candidates voted REGENERATE
5    - {finish_count} candidates voted FINISH
6
7    SYNTHESIS RULES:
8    1. Analyze critically and assume no candidate is fully correct
9    2. Prefer explicit checks over assumptions
10   3. Combine the best ideas from multiple candidates when appropriate
11   4. If candidates disagree fundamentally, choose the more robust approach
12   5. Combine best code ideas from REGENERATE candidates
13
14   OUTPUT FORMAT (strict):
15   - You may include brief reasoning first
16   - Then output "REGENERATE" on its own line followed by exactly ONE fenced code
     block, OR output "FINISH" on its own line
```

## H.4. Multi-turn prompt incentivizing debugging

We noticed that one failure case was inaccurate verification of task completion. Therefore, we experimented with a modified multi-turn prompt which incentivizes verification and debugging, however this did not empirically improve success rate.

| | Cube Lift | Cube Stack | Spill Wipe | Peg Insert | Cube Restack | Two Arm Lift | Two Arm Handover | Avg. |
|---|---|---|---|---|---|---|---|---|
| 3M | 97 | 98 | 100 | 0 | 89 | 74 | 20 | 68.29 |
| 3M + debug | 94 | 100 | 98 | 0 | 88 | 66 | 12 | 65.43 |

*Table 5.* Task completions between 3M and 3M + debug.

Modified multi-turn prompt:

```
1    You are acting as experienced debugger and task completion verifier. You will make
     no assumptions without explicit evidence.
2    Your task is to determine whether the program has actually completed the
     intended task, and to fix all bugs if any exist.
3    You can treat the observed differences between the current and previous state of
     the environment that are provided to you as a source of evidence, but they are not
     guaranteed to be accurate.
4
5    The following code blocks have been executed so far:
6    ```python
7    {executed_code}
8    ```
9    The current console stdout from the most recent code execution is:
10   ```
11   {console_stdout}
12   ```
13   The current console stderr from the most recent code execution is:
14   ```
15   {console_stderr}
16   ```
17
18   Do NOT trust printed outputs or logs blindly.
19
20   Proceed as follows to verify task completion:
21   1. Line-by-line inspect the executed code
22   2. Cross-check the code's behavior against the console stdout and stderr
23   3. Identify explicit, concrete evidence that each required step of the task was
     completed
24   4. Treat missing evidence, implicit assumptions, or partial signals as failure
25
26   Decision Rules:
27   - If the task is verifiably complete, respond with the single word 'FINISH'.
28   - If the task is not verifiably complete:
29     1. Identify specific bugs, failures, missing steps, or incorrect assumptions.
```

```
30        2. Explain why each issue prevents task completion, citing evidence from
       stdout, stderr, or code behavior.
31        3. If the same approach has been attempted multiple times without success, you
       MUST try a fundamentally different strategy.
32        4. Brainstorm multiple strategies to fix or verify these issues in the next
       code generation.
33        5. Synthesize your brainstorming into a single improved Python program that
       addresses all identified issues and verifiably completes the task.
34        6. Respond with the single word 'REGENERATE' followed immediately by new
       Python code in a fenced code block ('''python...''')
35
36     If you choose to regenerate code, you must include your reasoning process only
       AS COMMENTS at the start of the code explaining:
37     - What strategies/APIs have already been tried and why they were insufficient
38     - How your solution effectively addresses the identified issues
39
40     To reiterate, you must only respond with EXACTLY ONE of the following:
41     - The word 'FINISH' if you decide to stop generating code
42     - The word 'REGENERATE' followed immediately by new Python code in a SINGLE
       fenced code block ('''python...''').
```

## H.5. Model Ensemble Decreases Average Turn Count

We observed that applying a model ensemble for code generations decreases the average turn count. We noticed that M4 tends to retroactively implement bug fixes and API fallbacks, while the model ensemble preemptively anticipates failures beforehand, resulting in more robust code and lower turn count. See Appendix F.2.4 for a case study on retroactive vs preemptive behavior.

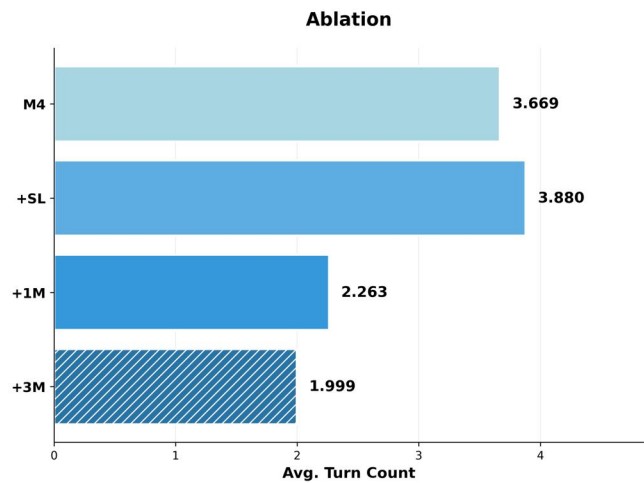

*Figure 23.* CaP-Agent0 decreases average turn count compared to M4

# I. Select Case Studies - Full Generated Code

## I.1. Stack These as High as You Can

Generated by Gemini-3-Pro as the coding agent and Gemini-3-Pro as the VDM agent.

```python
import numpy as np

def get_best_mask(masks):
    if not masks:
        return None
    return max(masks, key=lambda x: x["score"])

def get_object_metrics(mask, depth, intrinsics, extrinsics):
    """Returns center (x,y,z), top_z, bottom_z for a given object mask."""
    points = mask_to_world_points(mask, depth, intrinsics, extrinsics)
    if points.shape[0] == 0:
```

```
12          return None, None, None
13
14      # Get Oriented Bounding Box
15      bbox = get_oriented_bounding_box_from_3d_points(points)
16      center = bbox["center"]
17
18      # Calculate robust top and bottom Z from points
19      zs = points[:, 2]
20      if len(zs) == 0:
21          return None, None, None
22
23      top_z = np.percentile(zs, 98) # Use 98th percentile to filter outliers
24      bottom_z = np.percentile(zs, 2) # Use 2nd percentile
25
26      return center, top_z, bottom_z
27
28  def execute_pose_move(pose_matrix):
29      """Moves robot to the specified 4x4 pose matrix."""
30      pos, quat = decompose_transform(pose_matrix)
31      joints = solve_ik(pos, quat)
32      move_to_joints(joints)
33
34  def pick_and_place(obj_mask, target_center, target_top_z, depth, intrinsics, cam_pose):
35      """
36      Picks object defined by obj_mask and places it on target defined by target_center
        and target_top_z.
37      """
38      # 1. Analyze Object to be picked
39      obj_center, obj_top, obj_bottom = get_object_metrics(obj_mask, depth, intrinsics,
        cam_pose)
40      if obj_center is None:
41          print("Could not calculate object metrics.")
42          return
43
44      # 2. Plan Grasp
45      seg_map = obj_mask.astype(np.int32)
46      # plan_grasp returns poses in CAMERA FRAME
47      grasp_poses_cam, grasp_scores = plan_grasp(depth, intrinsics, seg_map)
48
49      if len(grasp_poses_cam) == 0:
50          print("No grasps generated by planner.")
51          return
52
53      # Select best top-down grasp.
54      # select_top_down_grasp takes grasps in CAMERA FRAME and returns the best in WORLD
        FRAME.
55      best_grasp_world, best_score = select_top_down_grasp(grasp_poses_cam, grasp_scores,
         cam_pose, vertical_threshold=0.8)
56
57      if best_grasp_world is None:
58          print("No strictly vertical grasp found. Trying relaxed threshold.")
59          best_grasp_world, best_score = select_top_down_grasp(grasp_poses_cam,
        grasp_scores, cam_pose, vertical_threshold=0.5)
60
61      if best_grasp_world is None:
62          print("No valid grasp found even with relaxed threshold.")
63          return
64
65      print(f"Executing grasp with score {best_score}")
66
67      # Calculate offset from TCP to bottom of object
68      # TCP Z at grasp - Object Bottom Z
69      grasp_z = best_grasp_world[2, 3]
70      tcp_to_bottom_offset = grasp_z - obj_bottom
71
72      # 3. Execute Pick
73      # Move to Pre-Grasp (Safety Hover)
74      pre_grasp = best_grasp_world.copy()
75      pre_grasp[2, 3] += 0.10
76      execute_pose_move(pre_grasp)
77
78      # Move to Grasp
79      execute_pose_move(best_grasp_world)
```

```
 80        close_gripper()
 81
 82        # Lift Up
 83        lift_pose = best_grasp_world.copy()
 84        lift_pose[2, 3] += 0.20
 85        execute_pose_move(lift_pose)
 86
 87        # 4. Execute Place
 88        # Calculate Place Pose
 89        place_pose = best_grasp_world.copy()
 90        place_pose[0, 3] = target_center[0]
 91        place_pose[1, 3] = target_center[1]
 92        # Target Z for TCP = Target Surface Z + Offset + Padding
 93        # Adding a small padding (1cm) to ensure we don't collide hard,
 94        # but close enough to drop safely.
 95        place_pose[2, 3] = target_top_z + tcp_to_bottom_offset + 0.015
 96
 97        # Move to Pre-Place (Safety Hover)
 98        pre_place = place_pose.copy()
 99        pre_place[2, 3] += 0.15
100        execute_pose_move(pre_place)
101
102        # Move to Place
103        execute_pose_move(place_pose)
104        open_gripper()
105
106        # Move to Post-Place (Lift Up)
107        execute_pose_move(pre_place)
108
109
110 # --- Main Execution ---
111
112 # 1. Reset
113 open_gripper()
114
115 # 2. Perception & Strategy: Stack Blue Cube on Yellow Cube
116 obs = get_observation()
117 rgb = obs["robot0_robotview"]["images"]["rgb"]
118 depth = obs["robot0_robotview"]["images"]["depth"]
119 intrinsics = obs["robot0_robotview"]["intrinsics"]
120 cam_pose = obs["robot0_robotview"]["pose_mat"]
121
122 # Find Yellow Cube (Base)
123 masks_yellow = segment_sam3_text_prompt(rgb, "yellow cube")
124 yellow_data = get_best_mask(masks_yellow)
125
126 # Find Blue Cube (Object 1)
127 masks_blue = segment_sam3_text_prompt(rgb, "blue cube")
128 blue_data = get_best_mask(masks_blue)
129
130 if yellow_data is not None and blue_data is not None:
131     # Get base metrics
132     yellow_center, yellow_top, _ = get_object_metrics(yellow_data["mask"], depth,
        intrinsics, cam_pose)
133
134     if yellow_center is not None:
135         print("Stacking Blue Cube on Yellow Cube...")
136         pick_and_place(blue_data["mask"], yellow_center, yellow_top, depth, intrinsics,
         cam_pose)
137     else:
138         print("Could not compute metrics for Yellow Cube")
139 else:
140     print("Could not find Blue or Yellow cube")
141
142
143 # 3. Perception & Strategy: Stack Ball on Blue Cube
144 # Re-observe because the blue cube has moved
145 obs = get_observation()
146 rgb = obs["robot0_robotview"]["images"]["rgb"]
147 depth = obs["robot0_robotview"]["images"]["depth"]
148 intrinsics = obs["robot0_robotview"]["intrinsics"]
149 cam_pose = obs["robot0_robotview"]["pose_mat"]
150
```

```
151  # Find Blue Cube again (New Base)
152  masks_blue_new = segment_sam3_text_prompt(rgb, "blue cube")
153  blue_data_new = get_best_mask(masks_blue_new)
154
155  # Find Tennis Ball (Object 2)
156  masks_ball = segment_sam3_text_prompt(rgb, "yellow tennis ball")
157  ball_data = get_best_mask(masks_ball)
158
159  if blue_data_new is not None and ball_data is not None:
160      # Get base metrics (blue cube is now the base)
161      blue_center_new, blue_top_new, _ = get_object_metrics(blue_data_new["mask"], depth,
          intrinsics, cam_pose)
162
163      if blue_center_new is not None:
164          print("Stacking Tennis Ball on Blue Cube...")
165          pick_and_place(ball_data["mask"], blue_center_new, blue_top_new, depth,
          intrinsics, cam_pose)
166      else:
167          print("Could not compute metrics for Blue Cube (new base)")
168  else:
169      print("Could not find Blue Cube or Tennis Ball for second stack")
```

## I.2. Place the Blue Cube on Top of the Yellow Cube

Generated by Gemini-3-Pro as the coding agent and Gemini-3-Pro as the VDM agent.

```python
1   import numpy as np
2
3   # --- Helper functions ---
4
5   def get_object_centroid(obs, text_prompt):
6       """
7       Finds the 3D centroid of an object specified by a text prompt.
8       Returns the centroid (x, y, z) in world frame and the mask.
9       """
10      rgb = obs["robot0_robotview"]["images"]["rgb"]
11      depth = obs["robot0_robotview"]["images"]["depth"]
12      intrinsics = obs["robot0_robotview"]["intrinsics"]
13      extrinsics = obs["robot0_robotview"]["pose_mat"]
14
15      # Segment object
16      results = segment_sam3_text_prompt(rgb, text_prompt)
17      if not results:
18          print(f"Object '{text_prompt}' not found.")
19          return None, None
20
21      best_result = max(results, key=lambda x: x["score"])
22      mask = best_result["mask"]
23
24      # Get 3D points
25      points = mask_to_world_points(mask, depth, intrinsics, extrinsics)
26      if points.shape[0] == 0:
27          print(f"No valid depth points for '{text_prompt}'.")
28          return None, None
29
30      centroid = np.mean(points, axis=0)
31      return centroid, mask
32
33  def pick_object(obs, text_prompt, z_offset=0.0):
34      """
35      Standard pick routine:
36      1. Segment object
37      2. Plan grasp
38      3. Execute pick trajectory (pre-grasp -> grasp -> close -> lift)
39      """
40      rgb = obs["robot0_robotview"]["images"]["rgb"]
41      depth = obs["robot0_robotview"]["images"]["depth"]
42      intrinsics = obs["robot0_robotview"]["intrinsics"]
43      extrinsics = obs["robot0_robotview"]["pose_mat"]
44
45      # 1. Segment
46      results = segment_sam3_text_prompt(rgb, text_prompt)
```

```
47     if not results:
48         print(f"Could not find {text_prompt} to pick.")
49         return False
50
51     # Get mask with highest score
52     best_res = max(results, key=lambda x: x["score"])
53     mask = best_res["mask"]
54
55     # 2. Plan Grasp
56     # Note: depth needs to be (H, W) or (H, W, 1) - API doc says (H, W) usually
       preferred for planner wrappers,
57     # but the function signature accepts (H, W, 1).
58     if depth.ndim == 3 and depth.shape[2] == 1:
59         depth_map = depth[:, :, 0]
60     else:
61         depth_map = depth
62
63     grasp_poses, grasp_scores = plan_grasp(depth_map, intrinsics, mask)
64
65     if len(grasp_scores) == 0:
66         print(f"No grasps found for {text_prompt}.")
67         return False
68
69     # Select best top-down grasp
70     best_pose, best_score = select_top_down_grasp(grasp_poses, grasp_scores, extrinsics
       )
71
72     if best_pose is None:
73         print("No valid top-down grasp found, falling back to highest score.")
74         best_idx = np.argmax(grasp_scores)
75         best_pose_cam = grasp_poses[best_idx]
76         best_pose = extrinsics @ best_pose_cam # Convert to world frame
77
78     # Decompose grasp pose
79     grasp_pos, grasp_quat = decompose_transform(best_pose)
80
81     # 3. Execute Pick
82     # Pre-grasp (hover 10cm above)
83     pre_grasp_pos = grasp_pos + np.array([0, 0, 0.1])
84
85     # Move to pre-grasp
86     joints = solve_ik(pre_grasp_pos, grasp_quat)
87     move_to_joints(joints)
88
89     # Open gripper
90     open_gripper()
91
92     # Move to grasp
93     joints = solve_ik(grasp_pos, grasp_quat)
94     move_to_joints(joints)
95
96     # Close gripper
97     close_gripper()
98
99     # Lift up (20cm)
100    lift_pos = grasp_pos + np.array([0, 0, 0.2])
101    joints = solve_ik(lift_pos, grasp_quat)
102    move_to_joints(joints)
103
104    return True
105
106 def place_at_position(position, height_offset=0.05):
107     """
108     Place currently held object at a specific world position.
109     Target orientation is usually top-down (gripper pointing down).
110     """
111     # Standard top-down orientation (gripper pointing -z)
112     # A common quaternion for top-down is pointing down z-axis.
113     # Let's assume the current grasp orientation is maintained or we define a fixed top
       -down.
114     # For simplicity, we often maintain the orientation we picked with, or reset to a
       known neutral top-down.
115     # Here, we will define a fixed top-down orientation [0, 1, 0, 0] (x-axis rotation
```

```
116        180 deg) or similar.
           # However, to be safe, let's just use the current robot configuration's orientation
            or a hardcoded one.
117        # Often [0, 1, 0, 0] is top down for Panda.
118        place_quat = np.array([0.0, 1.0, 0.0, 0.0])
119
120        target_pos = position + np.array([0, 0, height_offset])
121
122        # Move over target (high)
123        hover_pos = target_pos + np.array([0, 0, 0.15])
124        joints = solve_ik(hover_pos, place_quat)
125        move_to_joints(joints)
126
127        # Move down to place
128        joints = solve_ik(target_pos, place_quat)
129        move_to_joints(joints)
130
131        # Open gripper
132        open_gripper()
133
134        # Move back up
135        joints = solve_ik(hover_pos, place_quat)
136        move_to_joints(joints)
137
138    def main():
139        obs = get_observation()
140
141        # --- Strategy ---
142        # The stack is Green (top) -> Yellow (mid) -> Blue (bottom).
143        # Goal: Blue on Yellow.
144        # 1. Move Green to the side.
145        # 2. Move Yellow to another spot (target base).
146        # 3. Pick Blue.
147        # 4. Place Blue on top of Yellow.
148
149        # Defined drop zones relative to the table center or just hardcoded offsets based
           on workspace knowledge.
150        # Let's dynamically find a spot by looking at the initial stack position.
151
152        # Find the stack location (using the green cube at the top)
153        green_center, _ = get_object_centroid(obs, "green cube")
154        if green_center is None: return
155
156        # Define temporary drop zones relative to the stack
157        # Drop zone 1 (for green): 20cm to the left
158        drop_zone_green = green_center + np.array([-0.2, 0.0, -green_center[2] + 0.02]) # Z
            is roughly table height
159        # Drop zone 2 (for yellow): 20cm to the right (this will be the final destination
           for yellow)
160        drop_zone_yellow = green_center + np.array([0.2, 0.0, -green_center[2] + 0.02])
161
162        # --- Step 1: Remove Green Cube ---
163        print("Moving Green Cube...")
164        success = pick_object(obs, "green cube")
165        if success:
166            place_at_position(drop_zone_green, height_offset=0.05) # Place on table
167        else:
168            print("Failed to move green cube.")
169            return
170
171        # Refresh observation
172        obs = get_observation()
173
174        # --- Step 2: Move Yellow Cube ---
175        # Now Yellow should be exposed.
176        print("Moving Yellow Cube...")
177        success = pick_object(obs, "yellow cube")
178        if success:
179            # We place it at drop_zone_yellow. This is where we want to stack the blue one
           later.
180            place_at_position(drop_zone_yellow, height_offset=0.05)
181        else:
182            print("Failed to move yellow cube.")
```

```
183         return
184
185     # Refresh observation to find the new position of the yellow cube and the exposed
        blue cube
186     obs = get_observation()
187
188     # Get precise location of yellow cube now that it's moved
189     yellow_center, _ = get_object_centroid(obs, "yellow cube")
190     if yellow_center is None:
191         print("Lost track of yellow cube.")
192         return
193
194     # --- Step 3: Pick Blue Cube ---
195     print("Picking Blue Cube...")
196     success = pick_object(obs, "blue cube")
197     if not success:
198         print("Failed to pick blue cube.")
199         return
200
201     # --- Step 4: Place Blue on Yellow ---
202     print("Placing Blue on Yellow...")
203     # Target is yellow center, but offset Z by cube height (approx 5cm usually for
        these cubes)
204     # We add a small buffer.
205     cube_height_approx = 0.05
206     place_at_position(yellow_center, height_offset=cube_height_approx + 0.02)
207
208     # Done, gripper is already opened in place_at_position
209     print("Task completed.")
210
211 main()
```

### I.3. Additional Analysis Plots

We perform additional quantitative analysis on the cube stack task to understand the impact of multi-turn on success rate. We focus our analysis on Gemini 3 Pro (Google DeepMind, 2025), GPT 5.2 (OpenAI, 2025b), and Claude Opus 4.5 (Anthropic, 2025b), the three strongest performers on CaP-Bench. In particular, we plot the probability density function (PDF) of success over turns (Figure 24), PDF of success over tokens (Figure 25), multi-turn success rate vs turn for the cube stacking task in Figure 26. We find that for the cube stacking task, these models would perform the task on the first try, and use the subsequent turns for recovery. Successes for GPT 5.2 and Claude Opus 4.5 seem to have shorter code length than that of Gemini 3 Pro.

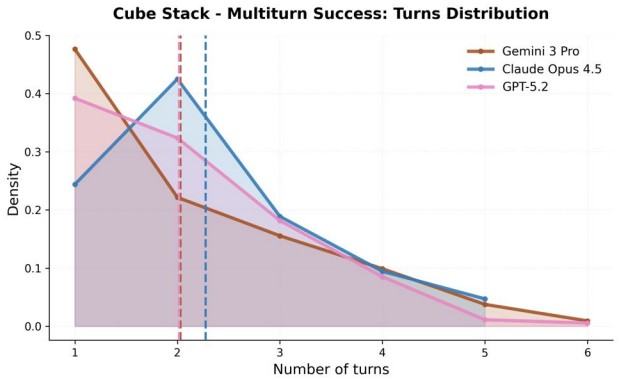

*Figure 24.* Distribution of multi-turn cube stack successes over turns. Results are averaged across M1, M2, and M3.

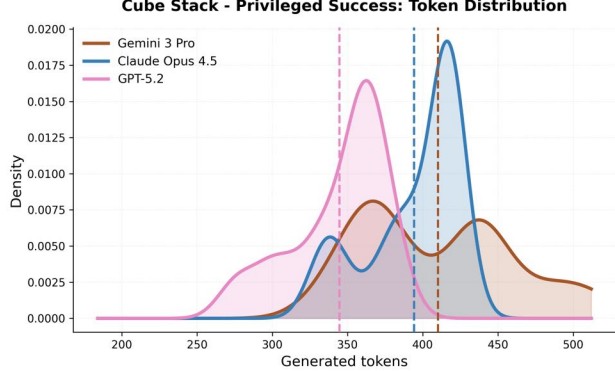

*Figure 25.* Distribution of cube stack successes (S1) over tokens

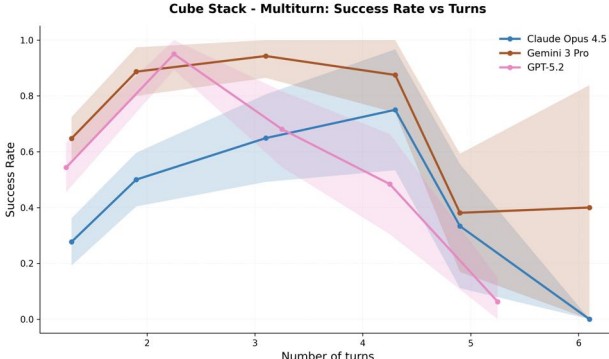

*Figure 26.* Cube stack multi-turn success rate vs. turns. Trials with too few or too many turns have lower success rate. Results are averaged across M1, M2, and M3.

## J. LIBERO-PRO Evaluation Results

We present detailed task-wise performance of OpenVLA, $\pi_0$, $\pi_{0.5}$, and CaP-Agent0 on LIBERO-PRO, which evaluates model generalization under initial position perturbations (Pos) and instruction perturbations (Task). Results are summarized in Table 6, Table 7 and Table 8. Each task was executed over 50 trials. In these tasks, CaP-Agent0 struggles with failures in perception, grasp generation, control APIs. For example, queries of "alphabet soup can" to SAM 3 often results in segmentations of the "tomato sauce can" also present in the scene. Also due to the occluded nature of these scenes and the camera not being top down, many grasps generated on desired objects will be at an angle, this, when combined with the lack of collision-aware motion planning often results in other objects in the scene being knocked over during execution, preventing a secure grasp on the desired object.

*Table 6.* Task-wise LIBERO-PRO performance of OpenVLA, $\pi_0$, $\pi_{0.5}$ and CaP-Agent0 on the **libero-object** benchmark. **Action notation:** $Place(obj, loc)$ = pick up object $obj$ and place $obj$ into/onto target $loc$.

| Task (Symbolic Form) | OpenVLA | | $\pi_0$ | | $\pi_{0.5}$ | | CaP-Agent0 | |
|---|---|---|---|---|---|---|---|---|
| | Pos | Task | Pos | Task | Pos | Task | Pos | Task |
| $Place$(alphabet_soup, basket) | 0.00 | 0.00 | 0.00 | 0.00 | 0.00 | 0.00 | 0.02 | 0.04 |
| $Place$(bbq_sauce, basket) | 0.00 | 0.00 | 0.00 | 0.00 | 1.00 | 0.02 | 0.12 | 0.42 |
| $Place$(butter, basket) | 0.00 | 0.00 | 0.00 | 0.00 | 0.54 | 0.00 | 0.26 | 0.18 |
| $Place$(chocolate_pudding, basket) | 0.00 | 0.00 | 0.00 | 0.00 | 0.00 | 0.02 | 0.18 | 0.48 |
| $Place$(cream_cheese, basket) | 0.00 | 0.00 | 0.10 | 0.00 | 0.00 | 0.00 | 0.12 | 0.06 |
| $Place$(ketchup, basket) | 0.00 | 0.00 | 0.00 | 0.00 | 0.20 | 0.02 | 0.32 | 0.12 |
| $Place$(milk, basket) | 0.00 | 0.00 | 0.00 | 0.00 | 0.00 | 0.00 | 0.38 | 0.02 |
| $Place$(orange_juice, basket) | 0.00 | 0.00 | 0.00 | 0.00 | 0.00 | 0.02 | 0.30 | 0.02 |
| $Place$(salad_dressing, basket) | 0.00 | 0.00 | 0.10 | 0.00 | 0.00 | 0.00 | 0.32 | 0.00 |
| $Place$(tomato_sauce, basket) | 0.00 | 0.00 | 0.00 | 0.00 | 0.00 | 0.00 | 0.16 | 0.48 |
| **Average** | 0.00 | 0.00 | 0.00 | 0.00 | 0.17 | 0.01 | **0.218** | **0.182** |

*Table 7.* Task-wise LIBERO-PRO performance of OpenVLA, $\pi_0$, $\pi_{0.5}$ and CaP-Agent0 on the **libero-goal** benchmark. **Action notation:** $Open(x, y)$ = open target $y$ of container $x$; $Put(obj, loc)$ = place object $obj$ onto/into location $loc$; $Push(obj, loc)$ = push object $obj$ toward location $loc$; $TurnOn(obj)$ = activate object $obj$.

| Task (Symbolic Form) | OpenVLA | | $\pi_0$ | | $\pi_{0.5}$ | | CaP-Agent0 | |
|---|---|---|---|---|---|---|---|---|
| | Pos | Task | Pos | Task | Pos | Task | Pos | Task |
| $Open(cabinet, drawer_{mid})$ | 0.00 | 0.00 | 0.00 | 0.00 | 0.00 | 0.04 | 0.00 | 0.00 |
| $Put(bowl, drawer_{top})$ | 0.00 | 0.00 | 0.00 | 0.00 | 0.94 | 0.02 | 0.04 | 0.00 |
| $Push(plate, stove_{front})$ | 0.00 | 0.00 | 0.00 | 0.00 | 0.00 | 0.00 | 0.00 | 0.10 |
| $Put(bowl, plate)$ | 0.00 | 0.00 | 0.00 | 0.00 | 0.00 | 0.02 | 0.36 | 0.38 |
| $Put(bowl, stove)$ | 0.00 | 0.00 | 0.00 | 0.00 | 0.00 | 0.04 | 0.22 | 0.12 |
| $Put(bowl, cabinet_{top})$ | 0.00 | 0.00 | 0.00 | 0.00 | 0.00 | 0.02 | 0.60 | 0.04 |
| $Put(cream\_cheese, bowl)$ | 0.00 | 0.00 | 0.00 | 0.00 | 0.98 | 0.02 | 0.04 | 0.34 |
| $Put(wine\_bottle, rack)$ | 0.00 | 0.00 | 0.00 | 0.00 | 0.88 | 0.02 | 0.02 | 0.12 |
| $Put(wine\_bottle, cabinet_{top})$ | 0.00 | 0.00 | 0.00 | 0.00 | 0.98 | 0.02 | 0.62 | 0.40 |
| $TurnOn(stove)$ | 0.00 | 0.00 | 0.00 | 0.00 | 0.00 | 0.00 | 0.66 | 0.18 |
| **Average** | 0.00 | 0.00 | 0.00 | 0.00 | **0.38** | 0.00 | 0.256 | **0.168** |

*Table 8.* Task-wise LIBERO-PRO performance of OpenVLA, $\pi_0$, $\pi_{0.5}$ and CaP-Agent0 on the **libero-spatial** benchmark. **Action notation:** $Pick(src, dst)$ = pick up object $bowl_{black}$ from location $src$ and place $bowl_{black}$ onto/into target $dst$.

| Task (Symbolic Form) | OpenVLA | | $\pi_0$ | | $\pi_{0.5}$ | | CaP-Agent0 | |
|---|---|---|---|---|---|---|---|---|
| | Pos | Task | Pos | Task | Pos | Task | Pos | Task |
| $Pick(between(plate, ramekin), plate)$ | 0.00 | 0.00 | 0.00 | 0.00 | 0.02 | 0.00 | 0.22 | 0.14 |
| $Pick(table\_center, plate)$ | 0.00 | 0.00 | 0.00 | 0.00 | 0.00 | 0.02 | 0.22 | 0.14 |
| $Pick(drawer_{top}(cabinet_{wood}), plate)$ | 0.00 | 0.00 | 0.00 | 0.00 | 0.00 | 0.00 | 0.02 | 0.10 |
| $Pick(next\_to(cookie\_box), plate)$ | 0.00 | 0.00 | 0.00 | 0.00 | 0.00 | 0.02 | 0.00 | 0.10 |
| $Pick(next\_to(plate), plate)$ | 0.00 | 0.00 | 0.00 | 0.00 | 0.00 | 0.00 | 0.10 | 0.20 |
| $Pick(next\_to(ramekin), plate)$ | 0.00 | 0.00 | 0.00 | 0.00 | 0.12 | 0.02 | 0.30 | 0.14 |
| $Pick(on(cookie\_box), plate)$ | 0.00 | 0.00 | 0.00 | 0.00 | 0.00 | 0.00 | 0.14 | 0.08 |
| $Pick(on(ramekin), plate)$ | 0.00 | 0.00 | 0.00 | 0.00 | 0.98 | 0.02 | 0.02 | 0.20 |
| $Pick(on(stove), plate)$ | 0.00 | 0.00 | 0.00 | 0.00 | 0.02 | 0.00 | 0.08 | 0.14 |
| $Pick(on(cabinet_{wood}), plate)$ | 0.00 | 0.00 | 0.00 | 0.00 | 0.90 | 0.00 | 0.08 | 0.16 |
| **Average** | 0.00 | 0.00 | 0.00 | 0.00 | **0.20** | 0.01 | 0.118 | **0.14** |

# K. Additional Clarifications

This section consolidates clarifications and added details requested by reviewers during the discussion period. It expands the protocol behind the human-expert baseline, the computational cost of CaP-Agent0, and the choice of VDM backbone.

### K.1. Human Expert Baseline Protocol

The human-expert curve in Figure 1 is *not* a single-shot human attempt. The baseline was written by a subset of the paper authors ($N=7$), each with $2+$ years of robotics programming experience. For each task and each tier, an author wrote a single Python script using exactly the same API primitives available to the model at that tier, and iterated through normal trial-and-error—reading execution traces, fixing bugs, and updating the script—until the program achieved high reliability. Human-written code uses the same primitives as the model; what differs is the iterative refinement loop performed offline by a human.

**Resulting upper bound.** The iterated human reference achieves $88.5\%$ average success on single-turn tiers, which we treat as a near-upper-bound for what is achievable when a robotics engineer hand-writes static code with full access to development-time iteration.

**Human effort budget.** The effort required to reach this near-upper-bound varies sharply with task complexity:

- **Simple pick-and-place tasks** (Cube Lift, Cube Stack): arrived at a working solution in a few hours (<1 day), with both low-level primitive and high-level code implementations, hill-climbed with a few rounds of trial-and-error.

- **Contact-rich and bimanual tasks** (Peg Insertion, Two-Arm Handover): identifying the right primitives and the right overall strategy took 2–3 weeks of iterative development per task. This estimate also includes the overhead of comparing alternative primitives and updating implementations as the underlying toolchain evolved (e.g., the Molmo $\rightarrow$ Molmo2 transition).

**Comparison to multi-turn agent recovery.** In contrast, CaP-Agent0's multi-turn recovery is fast at evaluation time: a complete evaluation of CaP-Agent0 on a Robosuite task takes approximately 2 minutes per trial (see Appendix K.2). The intended comparison is therefore between *static, single-turn code iterated on extensively by an experienced human* and an *interactive agent capable of online monitoring and recovery at deployment time*—which is precisely the practical gap that CaP-Agent0 is designed to address.

### K.2. Computational Cost Analysis

We report code-generation time and total trial time (LLM + execution) averaged over $N{=}20$ trials of cube stacking using Gemini-3-Pro. All measurements were collected on the same hardware with identical perception and control stacks.

| Tier | Avg LLM Code-Gen Time | Avg Trial Time |
|---|---|---|
| S1 (privileged) | 6.8 s | 12.9 s |
| S2 (non-privileged) | 9.6 s | 20.0 s |
| S3 (reduced API) | 23.8 s | 28.4 s |
| M1 (multi-turn, no VDM) | 20.1 s | 60.8 s |
| M3 (multi-turn + VDM) | 15.8 s | 113.6 s |

*Table 9.* Per-trial timing breakdown across CaP-Bench tiers (Gemini-3-Pro, cube stack, $N{=}20$).

**Comparison to VLAs.** For reference, $\pi_0$ (Black et al., 2024) reports $\sim$73 ms per action chunk and OpenVLA (Kim et al., 2024) reports $\sim$167 ms per action. VLAs are substantially faster per inference step because they emit single motor commands at high frequency. CaP-Agent0 operates at a different level of abstraction: each code-generation iteration (6.8–23.8 s) produces an *entire manipulation sequence*, not a single action. Direct latency comparison is therefore misleading; the relevant axis is *cost per task attempt* rather than *cost per action*.

### K.3. Visual Differencing Module: Model Choice

The Visual Differencing Module (VDM) used throughout the main paper is implemented with Gemini-3-Pro (Google DeepMind, 2025), which at the time of writing was the state-of-the-art on major multimodal benchmarks (MMMU-Pro: 81%; Video-MMMU: 87.6%). We choose the strongest available VLM to upper-bound the effect of language-grounded perception, ensuring that the observed performance gains in M3/M4 reflect the value of the *mechanism* (text grounding via an auxiliary VLM) rather than weakness of an under-powered captioner.

The VDM is *model-agnostic by design*: swapping the backbone requires a single configuration change in CaP-Gym, with no modifications to the coding-agent loop or to the rest of the benchmark. A systematic ablation across VDM backbones is an orthogonal axis of study that CaP-Bench supports out of the box and that we encourage the community to explore; the contribution of this paper is to demonstrate that VDM, as a *method*, materially helps multi-turn recovery.

