# OpenReview forum: "CaP-X: A Framework for Benchmarking and Improving Coding Agents for Robot Manipulation"
_ICML.cc/2026/Conference — ICML 2026 regular_

### Official Review · Reviewer_jYDf · 2026-03-10

**Soundness:** 3
**Presentation:** 3
**Significance:** 3
**Originality:** 3
**Overall Recommendation:** 5
**Confidence:** 4

**Summary:**

The paper introduces CaPX, an open-access framework designed to investigate the potential of "Code-as-Policy" in embodied robot manipulation. This includes CaP-Gym and CaP-Bench. The paper highlights a critical dependency on human-crafted abstractions for current models to function effectively, noting that performance suffers when these scaffolds are removed. To overcome these limitations, they propose CaP-Agent0, a training-free framework that achieves high reliability across simulated and real-world tasks, and CaP-RL, which demonstrates that incorporating verifiable rewards into reinforcement learning can further improve success rates and facilitate effective sim-to-real transfer.

**Compliance With Llm Reviewing Policy:**

Affirmed.

**Final Justification:**

I would like to thank the authors for their work in the rebuttal. My comments have been fully clarified. I would like to encourage the authors to clarify the aspects discussed during the rebuttal period in the paper. I updated my score accordingly.

**Key Questions For Authors:**

Please discuss points 1 and 3 from weaknesses.

**Limitations:**

yes

**Strengths And Weaknesses:**

Strengths
- The paper is generally well-written, and easy to follow.
- The field of study is also very relevant as it directly addresses the gap between high-level reasoning capabilities in Large Language Models and the low-level, high-precision control requirements of physical robotic manipulation. The authors provide a timely contribution that addresses a critical bottleneck in the current robotic landscape.
- The research offers a compelling and timely solution to the specific problem of developing embodied coding agents that can reliably operate without the "scaffolding" of human-engineered APIs. The authors introduce an effective approach that outperforms static human-written baselines and standard end-to-end Vision-Language-Action (VLA) models on multi-step manipulation tasks.
- The solution proposes not only the evaluation environment CaP-Bench, but also the CaP-Gym interactive coding framework, the training-free CaP-Agent0 architecture for multi-turn reasoning, and the CaP-RL paradigm for robust post-training of embodied agents.
- CaPX establishes a platform for advancing the development and evaluation of coding agents in robotics.

Weaknesses
- In the evaluation, the methodology of the baseline "Human" is not clear to me. The policy written by a Human can be imperfect, contain errors, which does not set a gold/static baseline. Beating a static human script using an interactive, self-correcting AI ensemble may be an unfair comparison that can  artificially inflate the impressiveness of the solution's success rate.
- Real-world experiments are limited and consist of simple stacking tasks, which do not fully capture the complexity of real-world environments. For that reason, the "minimal Sim-to-Real gap" claim should be further studied.
- The approach proposed in the paper does not consider a "recovery mechanism" for the cases in which there are errors in the Auto-Synthesized Skill Library. The agentic workflow can generate this library accumulating "spagetti code" or adding skills that work for some cases but not general enough to be implemented in any environment. If this happens the new task will be executed on top of faulty routines with no chance of recovery.
- As future work, it would be interesting to discuss if this ideas can be applied to dynamic environments, beyond the static ones considered in the paper.

---

> ### Author Rebuttal · Authors · 2026-03-30
>
> We thank the reviewer for recognizing CaP-X as a "compelling and timely solution." We address the reviewer's questions below:
>
> **W1 & Q1 (Human baseline fairness)**. We thank the reviewer for raising this concern and will revise to clarify: the human baseline is **not** a single-shot human attempt. It was written by a subset of the paper authors (N=7, each with 2+ years of robotics programming experience) using exactly the same API primitives available at each tier, with normal human iteration and debugging. Thus, the human programs use the same primitives as the model, but are manually refined to catch as many failure modes as possible. We view this as a near-upper-bound reference for the **single-turn setting**, which is precisely the level of performance we want single-turn models to achieve.
>
> The role of CaP-Agent0 is not intended to outperform the human baseline, but to address a practical limitation of static code. In real deployments, human roboticists can iterate extensively during development, but cannot stay in the loop indefinitely to monitor execution, detect new failure modes, and update the code online. Multi-turn interaction is designed to address this gap by allowing the agent to use feedback at test time to detect failures and generate corrective code. We will revise the paper to make this distinction clearer: the comparison is between **static single-turn code that's been iterated on by humans** and an **interactive agent capable of online monitoring and recovery**.
>
> **W2 (Real-world evaluation / sim-to-real)**. We agree with the reviewer that our current real-world results, while encouraging, are not sufficient to support a broad claim of a generally “minimal sim-to-real gap,” and we will soften this wording in the revision. Our intended claim is narrower: what transfers particularly well in our framework is **code as the action space**. In CaP-RL, the agent is trained to generate programs that compose a shared set of perception and control tools, while those tools themselves are fixed rather than jointly learned. As a result, the policy does not rely on simulator-specific end-to-end visual features, but instead learns to compose tool calls and symbolic operations shared across simulation and reality. We believe this shared code interface is an important reason for the encouraging transfer we observe, while agreeing that broader sim-to-real claims require more extensive real-world validation.
>
> We also clarify that the real-world evaluation goes beyond simple stacking. In addition to the Franka Panda sim-to-real experiments on cube lift and cube stack, Appendix A includes six zero-shot tasks on the AgiBot G1 robot: cluttered object retrieval, mechanical search, multimodal symbolic reasoning, learning from human feedback, physics-aware stacking, and a slightly longer-horizon elevator-button/navigation task. We will revise Section 4.3 to better distinguish these broader real-world demonstrations from the narrower CaP-RL sim-to-real experiments.
>
> **W3 & Q3 (Recovery from errors in the synthesized skill library)**. We thank the reviewer for raising this important concern (similar to reviewer j5qj), and we will revise to clarify the skill library generation process. The synthesized library in CaP-Agent0 is **not** an unconstrained accumulation of ad hoc code snippets. Instead, it is constructed from a verified pipeline: we first save code from ~2,876 successful S3-tier trials across all 12 models and 7 Robosuite tasks (i.e., not model-specific), then extract the functions used in these successful programs via regex, and finally prompt Gemini-3-Pro over this collected set of functions to synthesize **task-agnostic primitives** that are likely to be useful. This process yields a compact library of 9 verified task-agnostic functions (Appendix G.1). Moreover, the process is extensible: as more successful executions accumulate, the library can be periodically re-synthesized in an iterative manner. We will integrate these details into Section 4.2 to make the construction procedure more explicit.
>
> **W4 (Dynamic environments)**. As noted above, we’ll revise to emphasize that these experiments are intended as examples to kickstart community use of this benchmark, and we will add dynamic environments in future versions. We agree that extending these ideas to dynamic environments is an important and exciting future direction. The current paper focuses on relatively static settings because they provide a clean testbed for understanding abstraction, multi-turn recovery, and perceptual grounding. In the future works and conclusion section, we argued that “one promising direction is hybrid CaP+VLA policies, in which a coding agent manages high-level task logic and recovery while deferring low-level execution to VLA policies.” We encourage future works to explore such directions to address dynamic environments.

---

> > ### Author Rebuttal · Reviewer_jYDf · 2026-04-02
> >
> > I would like to thank the authors for the detailed response.
> >
> > Explicitly framing the contribution of CaP-Agent0 around its capacity for multi-turn online monitoring and recovery, rather than strictly outperforming this heavily refined static code, significantly strengthens the paper's practical motivation. I also appreciate the clarification regarding the verified pipeline used to construct the synthesized skill library. I encourage the authors to clarify these aspects in the camera-ready version of the manuscript.
> >
> > Regarding the human baseline, the clarification that it represents iterated code by experienced roboticists is very helpful. To better understand the practical advantage of the interactive agent, could the authors provide a rough estimate of the human effort required to achieve that near-upper-bound baseline compared to the time/token cost of the agent's multi-turn recovery?

---

> > > ### Author Response · Authors · 2026-04-03
> > >
> > > We thank Reviewer jYDf for the follow-up question and for the engagement with our rebuttal.
> > > > rough estimate of the human effort vs time for multi-turn
> > >
> > > The human effort required to construct the near-upper-bound baseline varies significantly by task complexity. For simpler tasks such as cube pick-and-place, we arrived at the code solution in a few hours (< 1 day, with both low-level primitive + high-level code implementation, and we hillclimb with a few rounds of trial-and-error). For more difficult tasks, such as peg insertion and bimanual handover, identifying the best primitives and overall strategy took 2–3 weeks of iterative development. This estimate also includes the overhead of comparing alternative primitives and updating the implementation as the underlying toolchain evolved (e.g., MolMo2 came out while we were exploring MolMo, and we ran ablation studies to confirm the effectiveness for MolMo2 and that there's no regression in performance). In contrast, multi-turn recovery at evaluation time is relatively fast: a complete evaluation for a RoboSuite task takes approximately 2 minutes (see the comparison table in our rebuttal to Reviewer j5qj). We will revise the manuscript to reflect this.

---

### Official Review · Reviewer_igP5 · 2026-03-13

**Soundness:** 3
**Presentation:** 4
**Significance:** 4
**Originality:** 4
**Overall Recommendation:** 5
**Confidence:** 4

**Summary:**

This paper presents CaP-X, a unified framework for systematically evaluating coding agents for robot manipulation. At the core of the framework is CaP-Gym, an interactive environment where agents control robots by generating and executing robotic programs that compose perception and control primitives. On top of this environment, the authors build CaP-Bench, which studies agent performance along three axes: abstraction level, multi-turn interaction, and perceptual grounding.

This paper shows how much current coding agents rely on human-crafted abstractions and other forms of scaffolding. The results show that agents perform well when these abstractions are available, but performance drops clearly as the interface becomes more low-level. This suggests that current coding agents still depend heavily on human-designed structure.

At the same time, the paper shows that increasing test-time computation can recover much of this gap. Based on these findings, the paper proposes CaP-Agent0, a strong training-free agent, and further extends the framework to CaP-RL, showing that reinforcement learning with verifiable rewards can improve performance and support sim2real transfer.

**Compliance With Llm Reviewing Policy:**

Affirmed.

**Final Justification:**

I believe this work has the potential to make a strong contribution to code-based agent research for physical robot control, and I look forward to seeing it released and trying it out in practice.

**Key Questions For Authors:**

Please see the weaknesses.

**Limitations:**

yes

**Strengths And Weaknesses:**

### Strengths:
The paper gives a clear and convincing structure to the research field between classical programming-based robot control and data-driven VLAs.
The paper introduces a broader and more reusable framework that connects the environment, the benchmark, a training-free improvement method, and an RL extension. This gives the work a broad scope and strong practical value.

The paper examines whether strong performance comes from the agent itself or from the high-level APIs provided by humans.
By analyzing both reduced abstraction and test-time scaffolding, the paper does more than report performance. It helps explain why the system works well in some settings and fails in others.
This benchmark-driven perspective makes the work particularly useful for future research.

### Weaknesses:
Although the integration of 39 tasks in CaP-Gym is an impressive infrastructure contribution, the main analysis in CaP-Bench is still centered on only 7 core tasks.
The experimental scale is substantial, since each task is evaluated for 100 trials per tier across 12 models, but the benchmark coverage itself still feels somewhat limited.

In that sense, CaP-Bench seems better suited for a controlled study of abstraction, multi-turn recovery, and perceptual grounding than for broad task coverage.
This design is well aligned with the authors’ analysis goals, but from the perspective of a long-term community benchmark, it leaves some open questions about generality and task coverage.

Given the likely community interest in CaP-Bench once it is released, it would be helpful for the authors to clarify the intended extensibility of the benchmark. In particular, can external users freely add new tasks and evaluation tiers within the CaP-Gym/CaP-Bench framework, or do the authors plan to maintain and periodically expand the benchmark with additional tasks over time?

---

> ### Author Rebuttal · Authors · 2026-03-30
>
> We thank the reviewer for the positive assessment and recognition of CaP-X's "broad scope and strong practical value" with "excellent" significance and originality.
>
> **(W1) Coverage of CaP-Bench**. As noted in response above, we will revise to clarify that the current CaP-Bench emphasizes depth rather than breadth. The 7 core table-top object manipulation tasks were intentionally chosen as a controlled subset to isolate the effects of abstraction level, multi-turn interaction, and perceptual grounding in a systematic evaluation setup, rather than to maximize task count. At the same time, upon release, CaP-Gym will support 187 tasks across Robosuite, LIBERO-PRO, and BEHAVIOR, including mobile manipulation tasks, and the paper already extends beyond the core 7-task study through CaP-Bench++, including 30 LIBERO-PRO tasks, 2 BEHAVIOR tasks, and additional real-world evaluations. We will revise the paper to make the distinction between the controlled core benchmark and the broader evaluation scope more explicit. Looking ahead, as CaP-Bench matures, we expect broader community evaluation to naturally expand toward CaP-Bench++ and the rest of CaP-Gym, and we are excited to continue monitoring progress over time.
>
> **(W2) Open-source design and extensibility**. We will revise to emphasize that CaP-X is intended to serve as an extensible long-term benchmark. The framework is modular and config-driven, allowing users to add new tasks, tiers, simulators, models, and perception backends with minimal changes to the surrounding codebase. New simulators only need to implement a standard environment interface, while benchmark tiers and API abstractions are designed to be swappable through configuration. This preserves comparability over time while still allowing the benchmark to grow with new tasks and new embodied models.

---

> > ### Author Rebuttal · Reviewer_igP5 · 2026-04-01
> >
> > I appreciate the authors’ thorough response to my concerns. The clarification around the benchmark’s design choices was helpful, and I hope the planned extensions are carried out as described.
> > In the revision, I encourage the authors to make the benchmark’s design intent and its user-friendly extensibility even more explicit for potential readers.
> >
> > Overall, I believe this work has the potential to make a strong contribution to code-based agent research for physical robot control, and I look forward to seeing it released and trying it out in practice.

---

### Official Review · Reviewer_j5qj · 2026-03-14

**Soundness:** 4
**Presentation:** 3
**Significance:** 3
**Originality:** 4
**Overall Recommendation:** 5
**Confidence:** 4

**Summary:**

This paper introduces CaP-X, a unified framework for systematically benchmarking and improving coding agents for robot control. The key motivation is that LLMs and VLMs can serve as general-purpose robot programmers to generate executable code that composes perception and control primitives for robotic manipulation (inherited from Code-as-Policy), but prior work overstated performance by relying on heavily hand-engineered abstractions. CaP-X consists of CaP-Gym, CaP-Bench, CaP-Agent0, and CaP-RL, enabling controlled evaluation across abstraction levels, interaction modes, and learning paradigms.

**Compliance With Llm Reviewing Policy:**

Affirmed.

**Final Justification:**

This paper is well-motivated and of high quality, and I hope the clarifications above will be incorporated into the revised version. I encourage the authors to fully release the code, models, and benchmark to benefit the research community of code-based robotic agents. I will raise my score to 5.

**Key Questions For Authors:**

- What exactly do “human expert solutions” mean? Are they human-written codes using the same primitives? Who are the human experts? How many people wrote the reference codes? Are they allowed to iterate or debug? Clarifying this (e.g., in Section 4.3 Takeaway 1) would help readers better understand the benchmark results.
- How many successful rollouts are collected to build the skill library? With which model? Are rollouts from all 7 tasks pooled together or handled separately? Which LLM performs the analysis step over the extracted functions? How often are skills collected? How are duplicate or near-duplicate functions handled?
- Which model is used for VDM? How does the choice of VDM model affect downstream performance? How frequently does VDM produce incorrect or hallucinated descriptions? How frequently do such errors cause task failures?
- What is the computational cost for CaP-Agent0? How does this compare to VLAs?

**Limitations:**

yes

**Strengths And Weaknesses:**

Strengths
- Well-structured benchmark design. Variables are carefully designed, and each tier differs from adjacent tiers by exactly one variable, enabling analysis of performance differences to specific factors. The zero-shot Pass@1 protocol is appropriate and realistic.
- Scale of evaluation. Experiments provide good insights spanning 12 frontier models across 7 tasks. Observations from the benchmark are insightful.
- Coherent system design. The designs of CaP-Agent0 and CaP-RL are directly motivated by observations from CaP-Bench, creating a strong connection from benchmark findings to system design.

Weaknesses
- Limited task coverage in the benchmark. Only 7 tasks are included in CaP-Bench, and they are all tabletop tasks with single or dual arms. The abstraction-level findings may not generalize to contact-rich or dexterous tasks.
- Insufficient justification for the three benchmark axes. The three axes are primarily introduced in Section 1 but are not sufficiently motivated in Section 4, where the benchmark is formally presented.
- Missing details of skill library. Skill library is a key component of CaP-Agent0 but lacks essential implementation details.
- Limited evaluation coverage in CaP-Bench++. The extended evaluation on LIBERO-PRO and BEHAVIOR uses only CaP-Agent0, without evaluating other models from CaP-Bench. BEHAVIOR evaluation covers only 2 tasks, which may not be enough for robust conclusions.
- Missing details of human expert solutions. The human expert baseline is the primary reference point for Takeaway 1 but is never described.
- No computational cost analysis. CaP-Agent0 makes multiple model calls per trial, but the paper does not report latency, token counts, API costs, inference time, etc.
- VDM quality is unanalyzed. The paper does not provide analysis of VDM failures, model choices of VDM, hallucination in VDM, and how VDM errors propagate to downstream task failure.

---

> ### Author Rebuttal · Authors · 2026-03-30
>
> We thank the reviewer for the thorough evaluation and recognition of the "well-structured benchmark design."  We’ll revise to emphasize that the reported results are intended as examples to kickstart community engagement with this open benchmark.
>
> **W1 (Task coverage)**: We include many tasks in the Appendix and will revise to encourage the community to expand evaluations with these environments and tasks using our open-source scripts.  CaP-Bench uses 7 core tasks for controlled ablation (7 tasks x 8 tiers x 12 models x 100 trials = 67,200 runs), but total evaluation is broader: 30 LIBERO-PRO tasks with comparisons between CaP-Agent0 and OpenVLA/pi0/pi0.5 (Table 2), Please NOTE: 2 BEHAVIOR mobile manipulation tasks (Table 3), quantitative evaluation on Franka for RL post-training (Table 4), and 6 real-world qualitative evaluation using the mobile robot AgiBot G1 (Appendix A.3, including occluded retrieval, symbolic reasoning, and elevator navigation). If accepted, CaP-Gym will integrates all 187 tasks (130 LIBERO-PRO + 7 Robosuite + 50 BEHAVIOR).
>
> **W2 (Benchmark axes)**: We will revise to specify: the three axes correspond to the primary design choices that practitioners face when deploying Code-as-Policy agents: (1) primitive abstraction: how much task structure to encode in APIs (eg, Code-as-Policies [Liang et al., 2023] uses high-level macros (such as put_first_on_second(’cyan block’, ’cyan bowl’)) vs. low-level primitives, i.e. segment_sam3_text_prompt(rgb, "red cube")); (2) level-of-iteration: zero-shot single-turn vs. multi-turn with structured execution feedback; (3) mode of grounding: feeding visual inputs directly into the coding agent vs. using an additional VLM to generate text descriptions. CaP-Bench isolates them via independently controllable tiers. We will add more explicit description and motivation in Section 4.
>
> **W3 (Skill library) & Q2**: We will revise to specify: generated code snippets from ~2,876 successful S3-tier trials across all 12 models and 7 robosuite tasks are saved (not model-specific). Then, all functions used in these code snippets are extracted by an LLM  (Gemini-3-Pro) via regex. Gemini-3-Pro is then prompted with all the functions to generate task-agnostic functions that it believes to be helpful. Result: 9 task-agnostic functions (Appendix G.1). The process can be made iterative, as more executions accumulate, the library can be re-synthesized. We will integrate these details into the skill library section 4.2.
>
> **W4 (CaP-Bench++)**: We will revise to specify: the purpose of introducing CaP-Bench++ is to compare coding agents with harnesses with 1) VLA and 2) human coders. Running all 12 models on LIBERO-PRO/BEHAVIOR across all 8 tiers of analysis requires enormous compute (BEHAVIOR needs Isaac Sim GPU-accelerated physics). We hope to maintain a public leaderboard and encourage the community to submit their agentic harnesses, integrate new APIs, simulators, and environments. The 2 BEHAVIOR tasks represent distinct challenges (table-height vs. floor-level, with navigation) and suggest CaP-Agent0 can exceed human baselines via multi-turn recovery (Table 3).
>
> **W5 & Q1 (Human experts)**: We will revise to specify: a subset of paper authors (N=7, each with 2+ years robotics programming experience) wrote single Python scripts using the same API primitives available at each tier, with iteration and debugging. Human-written code snippets use the same primitives as the model. The resulting code achieves near-upper-bound performance (88.5%) for single-turn tiers. We will add more details to the paper.
>
> **W6 & Q4 (Cost)**: We will revise to specify: code generation time measured per-trial (cube stack, Gemini-3-Pro, N=20):
>
> | Tier | Avg LLM Code Generation Time | Avg Trial Time |
> |------|-----------------|----------------|
> | S1 (privileged) | 6.8s | 12.9s |
> | S2 (non-privileged) | 9.6s | 20.0s |
> | S3 (reduced API) | 23.8s | 28.4s |
> | M1 (multi-turn, no VDM) | 20.1s | 60.8s |
> | M3 (multi-turn + VDM) | 15.8s | 113.6s |
>
> VLAs: we borrowed the following numbers from the papers: pi0 ~73ms/action chunk; OpenVLA 167ms/action (Kim et al., 2024). VLAs are faster per-action prediction, as they produce single motor commands at high frequency. However, we want to highlight that CaP-Agent0 operates at a different abstraction: each code generation iteration (6.8-23.8s) produces an entire manipulation sequence.
>
> **W7 & Q3 (VDM)**: We will revise to specify that we use Gemini-3-Pro since, at the time of writing the paper, it achieves state-of-the-art performance on major multimodal benchmarks MMMU-Pro (81%), Video-MMMU (87.6%). The VDM is model-agnostic by design (single config field swap to any VLM). A VDM model ablation is an orthogonal exercise that CaP-Bench supports, and we encourage future works to explore, but our contribution is to demonstrate that VDM, as a method, is helpful for multi-turn recovery.

---

> > ### Author Rebuttal · Reviewer_j5qj · 2026-04-04
> >
> > Thank the authors for the detailed response and clarifications. This paper is well-motivated and of high quality, and I hope the clarifications above will be incorporated into the revised version. I encourage the authors to fully release the code, models, and benchmark to benefit the research community of code-based robotic agents. I will raise my score to 5.

---

### Decision · Program_Chairs · 2026-04-30

**Decision:**

Accept (regular)

**Comment:**

The paper presents a well-structured and thoughtfully designed benchmark (CaP-Bench), where controlled variable differences enable precise analysis of performance factors under a realistic zero-shot Pass@1 setting. It evaluates 12 frontier models across 7 tasks, yielding broad and insightful findings. Building on these insights, the proposed systems (CaP-Agent0 and CaP-RL) are coherently motivated and tightly connected to the benchmark, forming a unified framework (CaPX) that integrates evaluation, training-free methods, and RL-based improvements.

Beyond performance, the work provides valuable analysis of whether gains stem from the agent itself or from human-designed APIs, offering a deeper understanding of system behavior across settings. The paper is clearly written and addresses a highly relevant challenge in bridging high-level reasoning in LLMs with low-level robotic control. Overall, it delivers a timely and practical contribution, introducing an effective framework and methods that advance embodied coding agents beyond reliance on human-engineered scaffolding.